# IoT-based demand-side energy management: Enhancing peak hour efficiency through automated control of appliances

Exaud Tweve [1,2], Godiana Philipo[1], Baraka Kichonge[1,3], Thomas Kivevele [1]*

1 School of Materials, Energy, Water and Environmental Sciences (MEWES), Nelson Mandela African Institution of Science and Technology, Arusha, Tanzania, 2 Electrical and Power Engineering Department, Mbeya University of Science and Technology, Mbeya, Tanzania, 3 Mechanical Engineering Department, Arusha Technical College, Arusha, Tanzania

* thomas.kivevele@nm-aist.ac.tz

## Abstract

Industrial solar microgrids experience pronounced peak demand due to aggregated production processes, auxiliary systems, and shift-based operation. Conventional demand response approaches rely on voluntary participation and aggregated energy metrics, which limits their effectiveness in industrial environments. This study presents an IoT-based demand-side energy management framework for industrial microgrids that explicitly distinguishes active power, reactive power, and apparent power during peak operation. The system integrates real-time power factor monitoring, automated reactive power compensation, and prioritized load control using low-latency IoT communication. A high-fidelity simulation model was developed in Proteus 8.15 and validated using a physical IoT prototype calibrated against industrial measurement data. Performance evaluation combined experimental measurements with historical load data from an operating industrial facility. Automatic power factor correction reduced apparent power demand by 41.58% due to reactive power mitigation. Real power decreased by 1.6 and 2.6% for both hardware and simulation respectively. The result shows that power factor correction reduces current and releases inverter capacity. It does not produce large reductions in real energy consumption. Automated shedding of non-critical loads reduced real power demand by 23.46% during peak periods. The non-critical loads include auxiliary lighting and support equipment rated at approximately 0.063 kW. These loads represent 4 percent of the monitored site load. Classification followed production continuity, safety requirements, and operational redundancy. Load factor increased by 23.4%, and both peak demand and peak-to-average ratio decreased by 20% relative to baseline operation. The results demonstrate that automated industrial demand-side energy management improves electrical performance and peak demand characteristics without reliance on voluntary user response. The proposed framework provides a practical foundation for power-quality-aware demand management in industrial solar microgrids.

**Data availability statement:** All relevant data are within the paper and its Supporting Information file.

**Funding:** The author(s) received no specific funding for this work.

**Competing interests:** The authors have declared that no competing interests exist.

# 1 Introduction

Global energy demand continues to increase due to population growth, industrial expansion, and rapid technological development [1,2]. Industrial energy consumption represents a significant share of total electricity use and places sustained pressure on generation and distribution infrastructure, particularly during peak operating hours [2]. In solar microgrids, inadequate coordination between supply and demand during peak periods leads to voltage instability, increased losses, higher operational costs, and in severe cases, system outages [3,4]. Conventional energy management strategies in microgrids focus primarily on generation-side optimization and centralized control [5,6]. These approaches often neglect end-user demand behavior and provide limited visibility into facility-level electrical performance. Manual monitoring and control of industrial loads remain common practice in many facilities, despite their susceptibility to human error, delayed response, and limited scalability [7,8]. These limitations motivate the adoption of automated demand-side energy management (DSEM) strategies.

Recent studies demonstrate the effectiveness of internet of things (IoT) technologies in enabling real-time monitoring and control of electrical loads [6,9]. IoT-based DSEM systems support continuous data acquisition, remote actuation, and real-time communication between consumers and microgrids. Existing implementations report reductions in peak demand and improvements in peak-to-average ratio through load scheduling and price-based demand response mechanisms [9,10]. However, most reported outcomes aggregate energy metrics without distinguishing between active power, reactive power, and apparent power, despite their distinct physical and operational implications [11–13]. In industrial environments, this lack of electrical granularity constrains the assessment of system performance. Reactive power demand increases current flow, raises conductor losses, and reduces available inverter capacity in solar microgrids [13,14]. Many IoT-based DSM studies focus exclusively on active power or energy consumption, overlooking the role of power factor correction in shaping demand profiles and improving capacity utilization. This gap limits the applicability of existing frameworks to industrial microgrids, where electrical performance margins are narrow and power quality requirements are strict.

Demand-side energy management strategies are commonly classified into price-based and incentive-based demand response programs [15,17]. Price-based schemes rely on dynamic tariffs to encourage consumers to shift load, while incentive-based approaches offer financial or non-financial rewards for demand reduction [18–23]. Both approaches depend on voluntary user participation and discretionary decision-making. In industrial facilities, such dependence reduces effectiveness during peak production periods, when operational continuity outweighs economic incentives [19,21,24]. Although individual industrial machines often operate under stable duty cycles, aggregated industrial demand exhibits significant temporal variability due to overlapping processes, auxiliary systems, and shift-based operation [25,26]. Measured data from industrial facilities show pronounced daily and monthly peaks, indicating clear opportunities for demand-side intervention at system level [27]. Effective industrial DSEM therefore requires automated control strategies that respond directly to real-time electrical conditions rather than user behavior.

This study addresses these limitations through an IoT-based DSEM framework tailored to industrial solar microgrids. The proposed system integrates real-time power factor monitoring, automated reactive power compensation, and prioritized load control within a unified architecture. The analysis explicitly separates active power, reactive power, and apparent power to evaluate their individual contributions to peak demand, load factor, and peak-to-average ratio. Unlike residential-focused DSM models, the framework targets industrial loads and emphasizes electrical performance rather than consumer price response. A high-fidelity simulation model is developed using Proteus 8.15 and validated through a physical IoT prototype calibrated against industrial measurement data. The system performance is evaluated using both experimental measurements and historical load data from an operating industrial facility. The results provide quantitative evidence of apparent power reduction through reactive power mitigation and real power reduction through automated non-critical load control. These outcomes demonstrate the technical feasibility of automated industrial DSEM and its relevance for improving power quality, reducing system stress, and enhancing solar microgrid operation.

## 2 Methodology

### 2.1 System architecture and design

This section defines the technical structure of the proposed energy management system and the logic guiding its configuration. The architecture addresses two persistent constraints at the end-user level in static single assignment (SSA): variability in solar generation and the operational impact of inductive and discretionary loads. The design integrates sensing, control, and communication into a single framework that supports demand prioritisation under constrained supply.

**2.1.1 Hardware components.** An architecture centred on direct demand control rather than passive monitoring as illustrated in Fig 1. The system separates critical from non-critical loads and treats power quality management and remote supervision as core functions. Table 1 summarises the technical specifications of the hardware components. The solar supply unit combines four 400 W photovoltaic panels, battery storage, and a single-phase inverter. This configuration

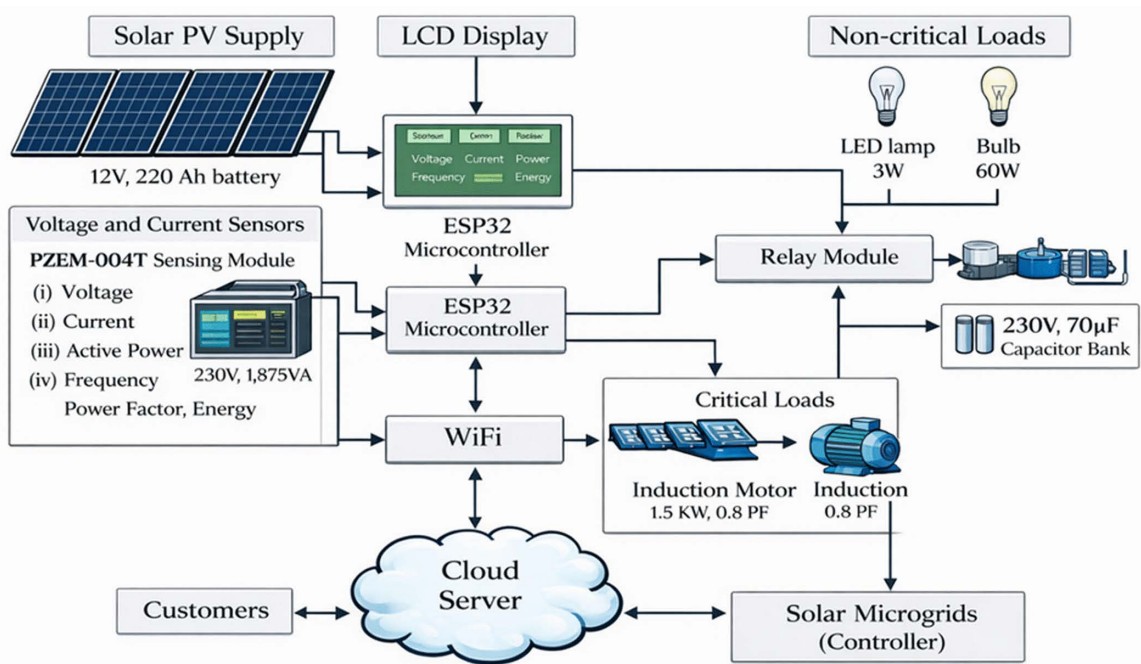

**Fig 1. IoT-based energy management system architecture.**

sustains base demand while tolerating short duration current surges from inductive loads. Battery capacity targets continuity of critical services during irradiance fluctuations. Inverter sizing reflects the need to accommodate motor starting currents without destabilising the AC bus. Electrical parameters are measured at the load interface using the PZEM-004T sensor. Active power, power factor, voltage, current, and frequency inform control decisions under constrained operating conditions. The ESP-32 microcontroller processes these inputs and executes load control through a relay module that enforces separation between critical and non-critical loads. Reactive power drawn by the induction motor is compensated locally through a capacitor bank. This arrangement stabilises the power factor and reduces apparent power demand at the inverter. A 20×4 liquid crystal display (LCD) provides local visibility of system status for operational oversight. Remote connectivity extends control through wireless fidelity (Wi-Fi) communication with a cloud server, enabling supervisory monitoring and remote load management within the same control structure.

**2.1.2 Communication interfaces.** The communication layer links local control to remote supervision under constrained bandwidth and power conditions. The system employs Message Queuing Telemetry Transport (MQTT), Hypertext Transfer Protocol Secure (HTTPS), and Wi-Fi to connect the microcontroller with the web server and user interface as in S1 Fig in S1 File for communication interfaces and control. MQTT manages continuous data exchange through a publish-subscribe structure that separates data producers from consumers. This structure supports stable operation for constrained controllers, while the broker controls message routing and access through credential-based authentication. HTTPS supports secure transactions that require data integrity and controlled access, including user interaction and configuration exchange. Transport Layer Security protects data confidentiality during transmission between the controller and the server. Wi-Fi provides the local network connection for bidirectional data flow, supporting real-time status updates and control actions. A web-based interface exposes system status and control functions, treating communication as an integral part of operational control rather than a standalone monitoring layer.

## 2.2 Simulation environment and setup

The simulation examines operational behavior under controlled conditions and mirrors the functional structure of the proposed system. The analysis targets voltage and current response, power factor variation, and relay actuation during load transitions. The objective is behavioral consistency with observed system responses under comparable loading, while experimental evaluation remains addressed elsewhere. Fig 2 depicts the simulated closed loop, integrating sensing, control, actuation, and communication within a single environment. The figure represents the simulation implementation rather than an abstract system view.

**Table 1. Technical specifications of hardware components.**

| S/N | Components | Technical specifications |
|---|---|---|
| 1. | Solar panel | 12 Vdc, 450W, operating voltage 41 V, operating current 10.5 A, open circuit voltage 48 V, short circuit current 11.5 A |
| 2. | Battery | Lithium-ion battery 12 Vdc, 220Ah, |
| 3. | Inverter | 230V, 2,000 VA, efficiency 90% |
| 4 | ESP-32 | 3.3 Vdc, processor 24 MHz, Flash memory: 4 MB. |
| 5. | LCD | 20x4 display, input voltage 3Vdc, ambient temperature 40°C. |
| 6. | PZEM-004T sensor | 230V,100A, accuracy $\pm 0.5$, Active power measuring range 0-23kW, PF range 0–1, and frequency measuring range 45 Hz −65 Hz. |
| 7. | Capacitor | 230V, 70$\mu$F. |
| 8. | Induction motor (critical load) | Single phase 230V, 8A, 1.5 kW, 0.8 PF lagging. |
| 9. | Two bulbs (non-critical loads) | LED lamp 230 V, 0.1 A, 3 W, 50 Hz and bulb 230 V, 0.26 A, 60 W, 50 Hz. |

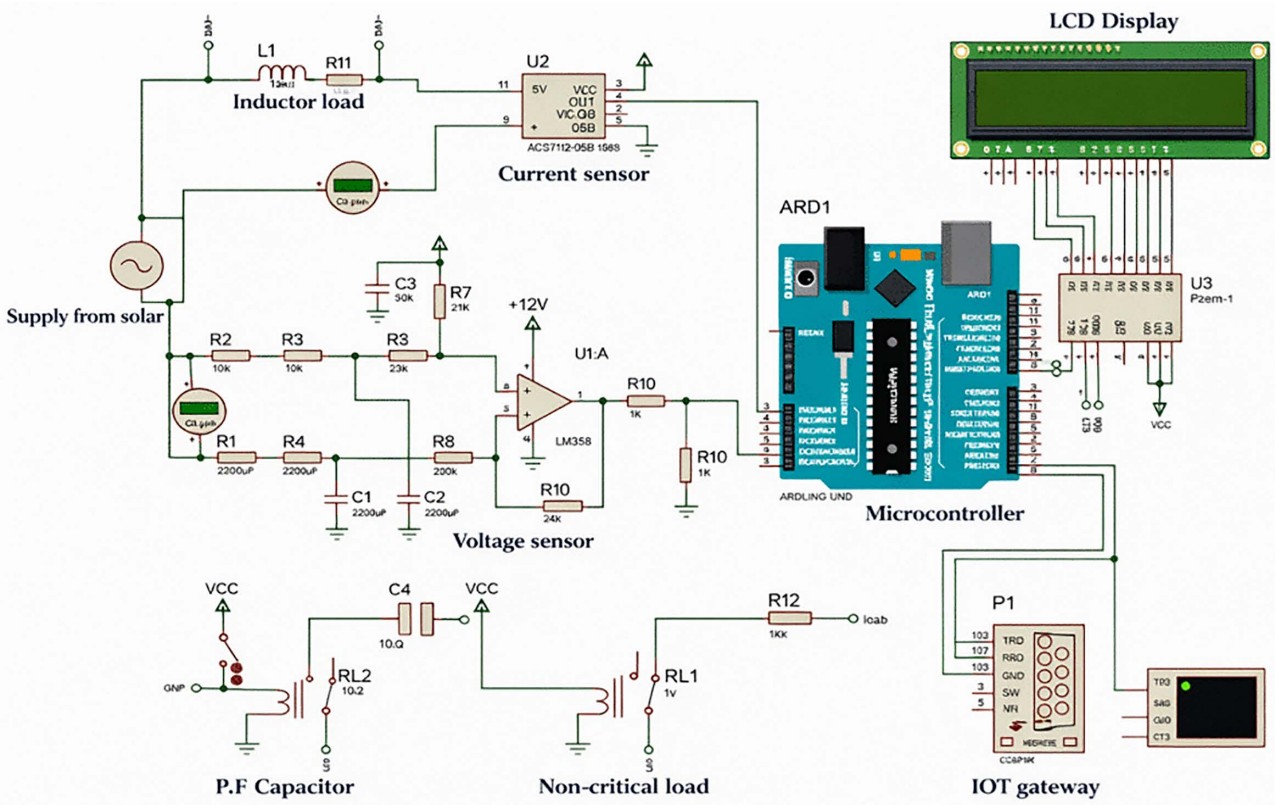

**Fig 2. Remote control of electrical energy consumption at the end users.**

**2.2.1 Simulation tools and software.** The system is implemented in Proteus 8.15 Professional, which supports mixed signal circuit simulation with Arduino-based controllers. An Arduino Uno R3 serves as the control unit and executes firmware developed in the Arduino integrated development environment (IDE) using standard C++ libraries. Simulated voltage and current sensing circuits provide real time inputs to the controller. The control logic applies fixed thresholds to power factor and real power. One relay connects the capacitor bank when the power factor drops below 0.95, while a second relay disconnects non-critical loads during predefined peak periods. Switching occurs only after sustained threshold violation over a defined sampling interval, which suppresses relay oscillation and stabilises operation. Measured parameters appear on a simulated LCD and transfer to a server through the Proteus Communications Peripheral Interface Model acting as an IoT gateway. The server processes incoming data for web-based display and returns control commands to the simulated circuit. This closed-loop arrangement reflects the functional scope shown in Fig 2 and supports real-time monitoring and remote load control within the limits of the simulation.

**2.2.2 Simulation flow and control logic.** The control strategy adopts a deterministic, rule-based structure aimed at stable operation under variable load conditions. Each decision relies on a fixed number of arithmetic comparisons and logical evaluations, so computational demand remains invariant as the number of connected loads increases. The control logic applies deterministic thresholds to evaluate power factor and real power in real time. The controller switches capacitor banks only after consecutive power factor readings remain below 0.95 for a defined observation window. The controller disconnects non-critical loads when real power exceeds the defined peak threshold. System status appears on the local LCD and transmits to the cloud server for remote access, while sustained abnormal consumption triggers user

notification via electronic mail. Fig 3 illustrates the corresponding control flow proofs. The choice of control framework balances simplicity and robustness to uncertainty. A rule-based deterministic strategy works well when demand patterns and tariffs are predictable, using predefined logic triggered by load thresholds or time-of-use signals. It is simple to implement, computationally efficient for IoT devices, and easy to interpret, making it suitable for initial deployment in structured environments. The proposed control strategy follows a rule-based deterministic structure. Fixed thresholds act as operational risk limits. This approach fits industrial systems with stable load behavior and strict production constraints.

However, real-world demand-side energy systems face uncertainty from user behavior, renewable generation, and network constraints. In such cases, a risk-aware learning-based policy performs better by treating control as a sequential decision problem under uncertainty. By incorporating predictive models and risk measures like Conditional Value-at-Risk (CVaR) or probabilistic peak constraints, it optimizes peak-hour efficiency while limiting extreme demand events. Reinforcement learning approaches also allow continuous adaptation to changing consumption patterns without requiring

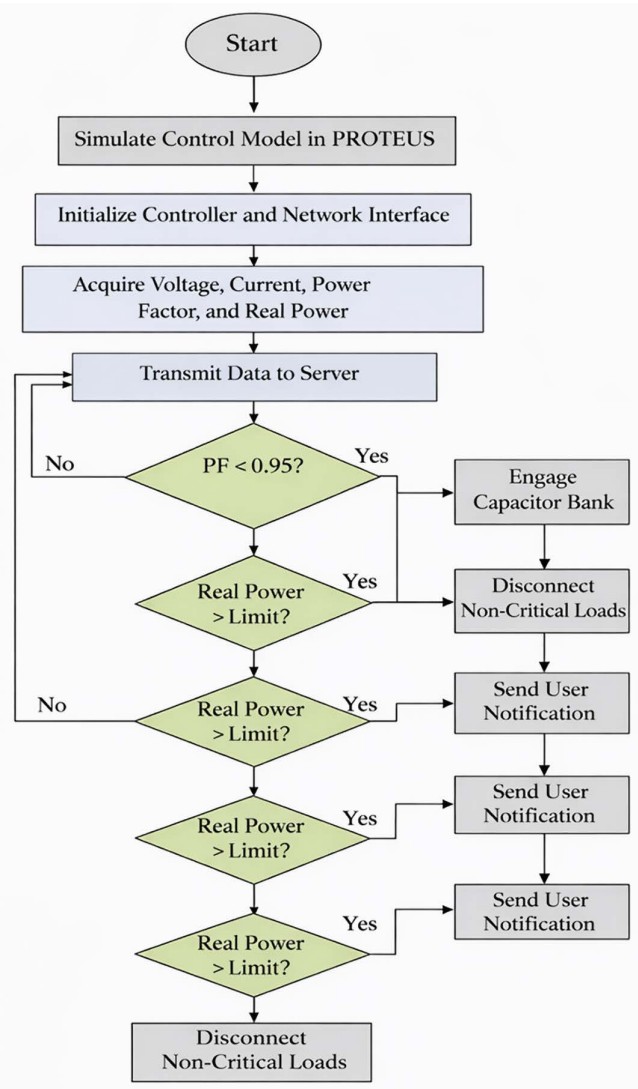

**Fig 3. Flow chart diagram of the research methodology.**

 

system re-modeling. Risk-aware or learning-based policies suit systems with stochastic demand and flexible operation but require higher computational effort. Therefore, while rule-based control offers robustness and ease of deployment in deterministic settings, a risk-aware learning-based framework is more suitable for dynamic IoT-enabled demand-side management systems where uncertainty quantification, adaptive optimization, and probabilistic peak mitigation are essential for sustained performance improvement. System stability is assessed through sustained operation without oscillatory switching during simulation and prototype testing, rather than through formal analytical.

**2.2.3 Data transmission and remote monitoring.** During experimental operation, the system transmitted MQTT data updates at fixed sampling intervals rate of 0.33 Hz (3-second transmission interval) to optimize broker load, network bandwidth usage, and end-to-end latency. Measurement data from the meter were encoded in JavaScript Object Notation (JSON) format prior to publication to the broker. Performance evaluation during experimental testing indicated an average MQTT round-trip latency of 800 milliseconds, while the controller executed control actions locally upon threshold violation. The study did not target detailed network latency benchmarking. Observed communication delays did not interfere with local control execution. These observations motivate future work on edge-level control to further improve responsiveness during peak operation. This web application enables users to remotely issue commands, such as activating capacitors or deactivating non-critical loads within the Proteus circuit model. With this setup, Suppliers/users can efficiently monitor and control energy consumption, enhancing overall management and sustainability of power. The source code for Node. js in simulation is hosted on the Github repository links simulation web app - https://github.com/Exaud1996/microgrid_simulation.git. Fig 4 designed webpage for smart electrical energy management system. It presents the monitoring dashboard used during simulation, showing load status and power factor conditions.

## 2.3 Hardware prototype implementation

The hardware prototype examines system behavior under physical operating conditions and evaluates practical feasibility beyond simulation. The assessment focuses on control stability, measurement reliability, and consistency of response under variable loading.

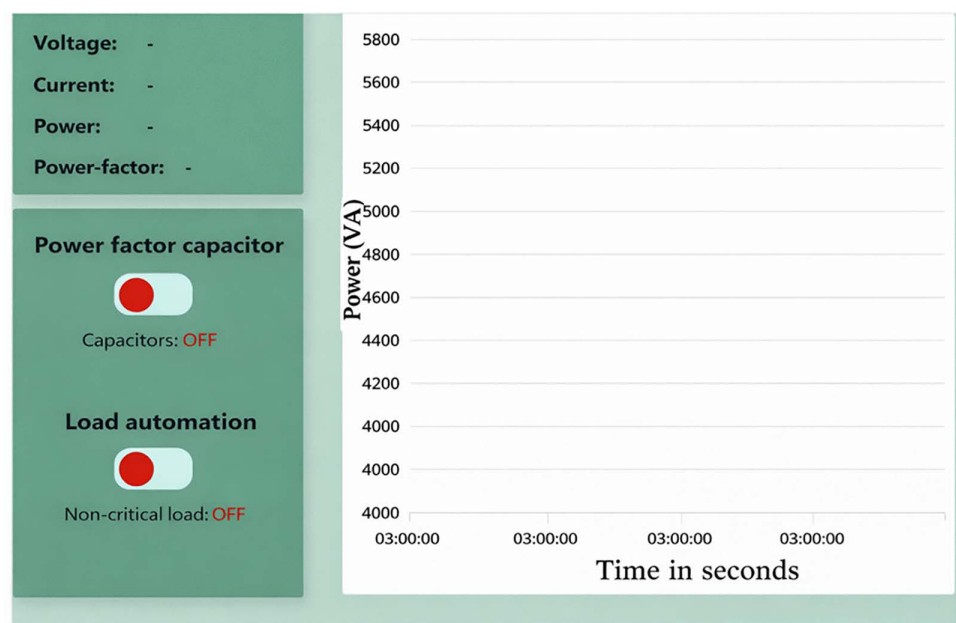

**Fig 4. Webpage for smart electrical energy management system.**

**2.3.1 Prototype design and schematic diagram.** The prototype integrates an ESP-32 microcontroller, PZEM-004T sensors, relay modules, and a local display in a controlled test setup. The experiment began by evaluating the accuracy of electrical parameter measurements, namely voltage, current, and power factor without reactive power compensation. The MS2203 was used as the reference instrument for comparison with the PZEM-004T. Sensor calibration against a reference digital clamp meter (MS2203) achieved measurement accuracy within ±1.5 percent. This step establishes confidence in the data driving control actions. Table 2 indicates values of the electrical parameters measured by MS2203 against PZEM-004T.

Testing covered three operating conditions. The baseline case assessed reactive power demand without compensation. The power factor correction case engaged the capacitor bank when the power factor dropped below 0.95, reducing apparent power demand. The load shedding case disconnected non-critical loads during peak demand periods through remote commands. Observed trends aligned with established demand-side management outcomes reported in price-based studies, despite differences in data sources. Fig 5 shows the schematic diagram of the prototype hardware. The ESP-32 serves as the central controller, processing voltage and current measurements from the PZEM-004T sensor and presenting key parameters on an LCD. A rectified AC supply powers the control circuitry. A four-channel relay switches connected loads. The monitored equipment includes a single-phase induction motor classified as a critical load and two lighting units treated as non-critical loads. Power factor correction targets the motor load, while load shedding governs lighting demand. This configuration reflects the priorities defined in the simulation and supports end-user energy management under constrained supply condition.

**2.3.2 System integration and testing.** System integration links sensing, control, actuation, and communication into a single operational chain. The prototype operates on a 230 V AC supply derived from the solar source and incorporates an ESP-32 controller, a PZEM-004T sensor, a 20 × 4 LCD, a power factor correction capacitor, and segregated critical and non-critical loads. The single-phase induction motor represents the critical load, while two lighting units represent discretionary demand. The assembled prototype used for monitoring and control as shown in Fig 6.

The ESP-32, sensing module, display, and relay hardware reside within a dedicated control enclosure, shown in Fig 7 the control box and its main components. The controller executes all measurement processing, control decisions, and communication tasks locally. The PZEM-004T sensor supplies voltage, current, real power, energy, frequency, and power factor measurements through a serial interface. These parameters appear on the LCD to support local inspection and fault identification. The hardware layout follows the same functional separation applied in the simulation, which preserves consistency between virtual and physical operation.

Remote supervision occurs through a web-based interface, illustrated in Fig 8. The server runs on Node.js with the Express framework and receives device data through an MQTT broker. Incoming data streams update system status on the interface without interfering with local control execution. Control actions issued remotely invoke predefined switching operations and do not modify controller logic. Cloud deployment isolates supervisory access from field-level control and supports continuous availability. Containerised deployment standardises the runtime environment and limits configuration variation. Through its GitHub repository, anyone can access, modify, and contribute to its development. Whether looking

**Table 2. Measurement of electrical parameters comparing MS2203 against PZEM-004T.**

| Measurement | MS2203 | | | PZEM-004T | | |
|---|---|---|---|---|---|---|
| | Voltage (V) | Current (A) | Power Factor | Voltage (V) | Current (A) | Power Factor |
| Data 1 | 230 | 5 | 0.85 | 233.6 | 5.1 | 0.86 |
| Data 2 | 232 | 4.8 | 0.89 | 235.5 | 4.9 | 0.91 |
| Data 3 | 231.5 | 5.2 | 0.83 | 228.4 | 5.4 | 0.82 |
| Data 4 | 230.8 | 4.7 | 0.9 | 234 | 4.6 | 0.93 |
| Data 5 | 232.7 | 4.5 | 0.87 | 229 | 4.4 | 0.89 |

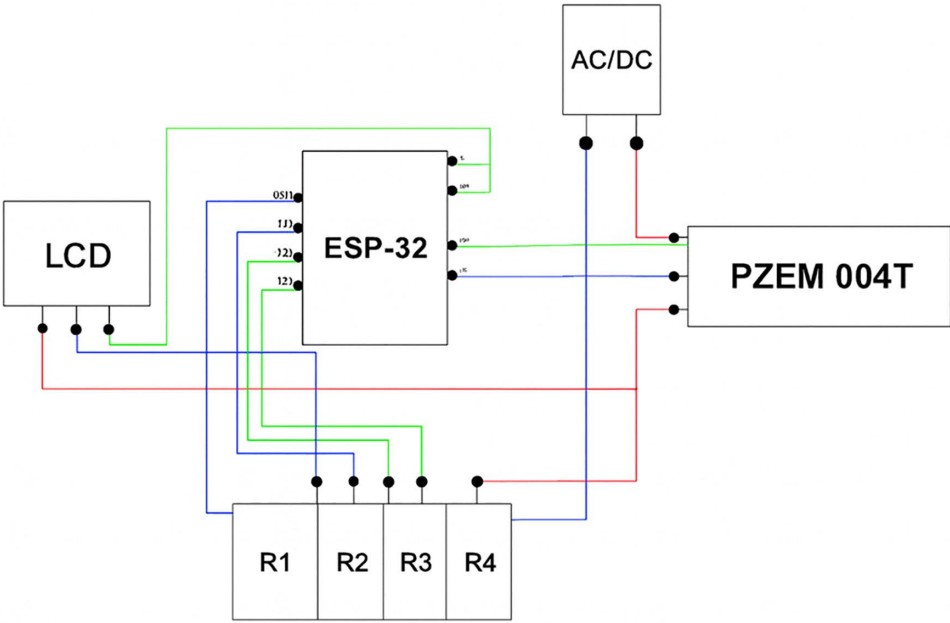

**Fig 5. Schematic diagram for load control.**

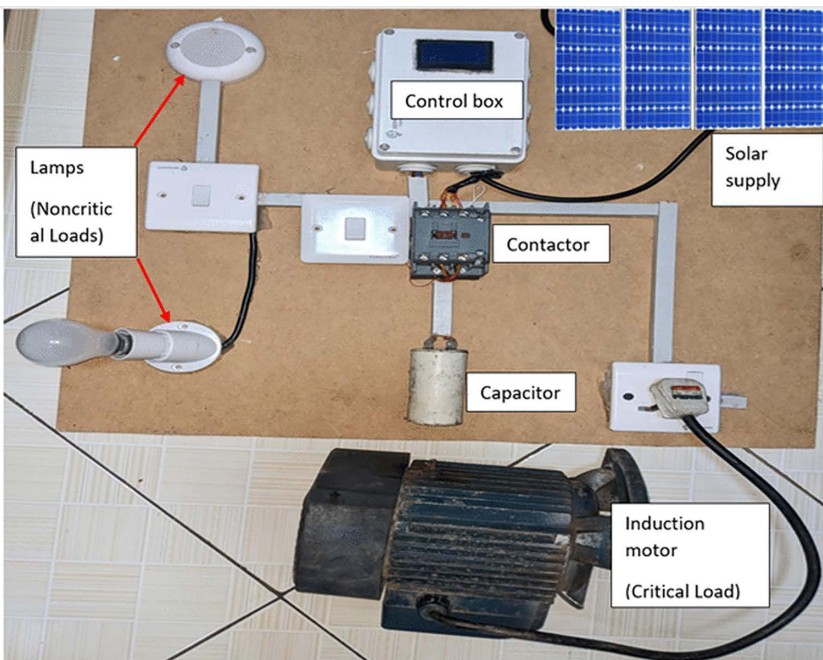

**Fig 6. Prototype for monitoring of equipment and appliances.**

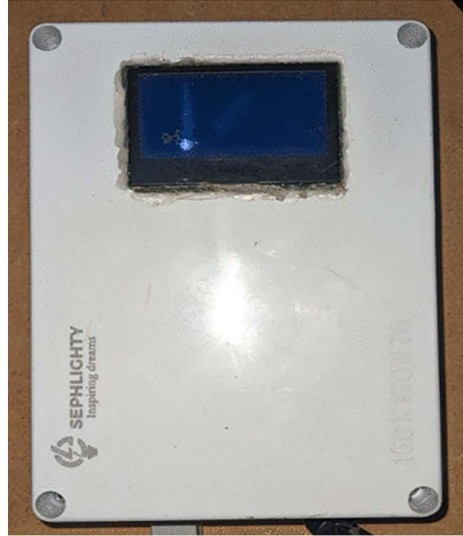
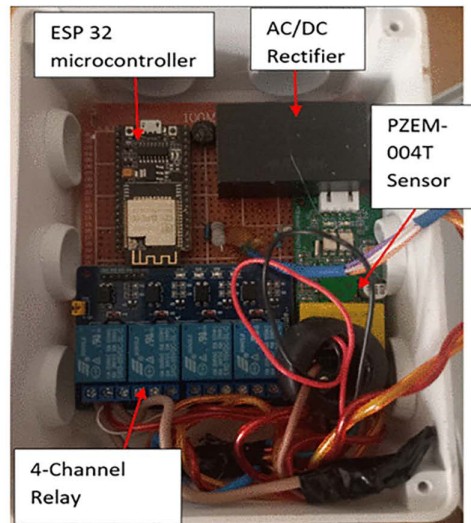

**Fig 7. Control box and its main components.**

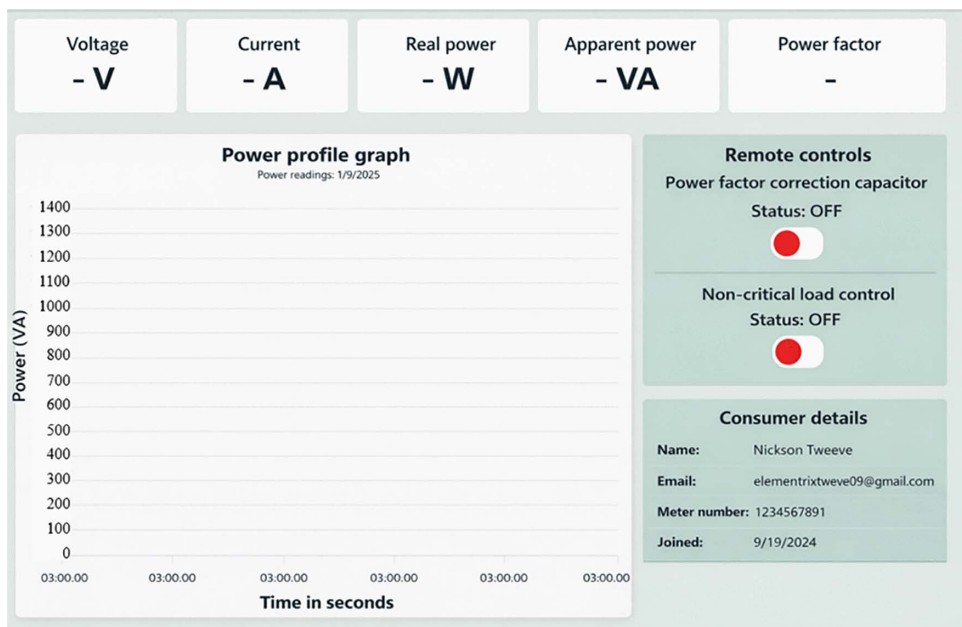

**Fig 8. A webpage for remote control of power factor and noncritical loads.**

for exploring the source code, make improvements, or build the version of Node.js, the open-source nature of the project supports wide-ranging innovation and collaboration. Therefore, the source code for Node.js in hardware for our study is hosted on the Github repository links namely:

Web app - https://github.com/Exaud1996/microgrid.git

Hardware - https://github.com/Exaud1996/microgrid_hardware.git

Prototype operation assumes adequate server capacity, stable network connectivity at the deployment site, and user access to internet-enabled devices. These conditions bound the evaluation to control performance and integration behavior rather than to communication infrastructure reliability.

### 2.4 Data collection and experimental setup

**2.4.1 Experimental site and instrumentation.** Data collection was carried out for a period of 2 months from February 1st to March 31st, 2023. During this activity, the reading of energy consumption from the Digital power clamp on digital meter (MS2203) and installed digital meters were recorded. Digital power clamp on digital meter (MS2203) implements handheld feature structure design that is appropriate for testing and maintenance of electrical systems and power equipment and appliances. This instrument is selected for measurement because it has steady performance, and high accuracy and incorporates RS232 for computer and other devices interface. The measurement was carried out to observe the relationship of power consumption between the supply and demand of the solar microgrids. The energy and power factor from both demand and supply is useful to study the energy management and energy equilibrium between supply and demand.

**2.4.2 Data acquisition process.** The digital power clamp on digital meter (MS2203) has temporary memory (data hold function) which stores the last measured value on the screen until the user resets it. The meter was used to measure the active power, reactive power, apparent power, energy, and power factor as well as the voltage and current. Furthermore, the measurement of these parameters were done on an hourly basis to investigate the energy consumption on a daily profile. The measured values were saved and uploaded to the computer for further analysis. The readings of energy consumption from digital energy meters of critical and noncritical loads and historical data were downloaded from the database system of Tanzania breweries limited (TBL) Mbeya plant, at the brewery section. The brewery section is subdivided into eight subsections namely; Malt Tower, Beer process cellars, Yeast and Filter cellars, Fermentation and storage vessels, Brew house Auxiliaries, Malt Tower, Motor control Centre (MCC), and Brew house office. Each subsection has a digital energy meter that records total energy consumption, with readings manually collected by an operator daily. The energy consumption for each subsection is recorded on a daily, weekly, monthly, and yearly basis. The data collected from the digital energy meter are uploaded and utilized in load profiles within DSEM to enhance power quality, improve energy efficiency, and optimize solar microgrid management. Furthermore, the loads under this section are classified into the critical and noncritical loads. The classification as critical or non-critical is governed by factors such as safety, operational necessity, regulatory requirements, and redundancy. Critical loads are systems essential to maintaining production flow while non-critical loads are loads auxiliary or support systems that do not halt production if they fail. Each type of load has a digital meter that records total energy consumed and the energy generated from the solar microgrid. In addition, power factor meters are installed to monitor the power factor of these loads.

**2.4.3 Load profiling and metering.** The load profiles for the brewery section based on Fig 9 is the Brewery daily load profile for the first day of March 2023 and Fig 10 is the Brewery monthly load profile for March 2023. The data for Figs 9 and 10 are as in S2 Table and S3 Table in S1 File respectively. Fig 9 shows the highest energy consumption of 150 kWh occurs at hour 11 and the average peak energy consumption is 143.3 kWh/h occurs at the period ranges at the operating window from 09:30–12:30. This value represents energy consumed over a three-hour period. It does not represent instantaneous power demand. During this period, energy consumption is at its peak due to the regular production processes, as well as additional activities such as workshop operations and keg cleaning using air compressors. This situation results in stress on the supply of solar microgrids. It signifies that energy management techniques such as direct load control need to be implemented to reduce energy consumption. In Fig 10 the peak energy of 1013 kWh occurs on 5th March, 2023. Other maximum energy observed are 812 kWh, 768 kWh, 694 kWh, 689 kWh, and 716 kWh occurring on 3rd, 12th, 20th, 26th, and 30th March respectively. These values depict that a higher current is drawn from the supply, therefore power factor improvement and load shedding methods of energy minimization for noncritical loads are to be applied. The daily and monthly load factors are 0.47 and 0.53 respectively, according to equation (1) assuming 1-hour

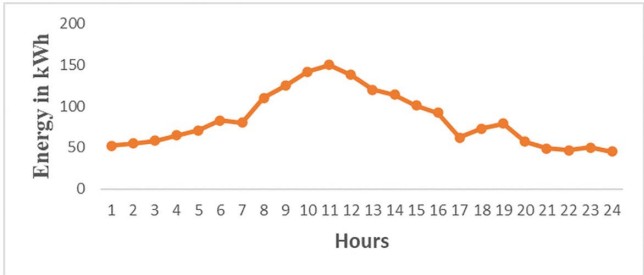

**Fig 9. Brewery daily energy profile for the first day of March 2023.**

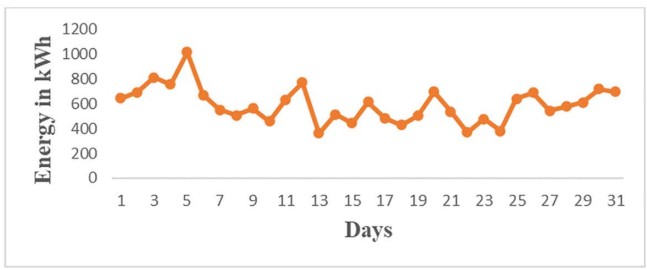

**Fig 10. Brewery monthly energy profile for March 2023.**

intervals. These values are low because the recommended load factor should be closer to 1 for flattening the load curves [25]. Therefore, the load factor (LF) can be enhanced by implementing an effective operational plan that integrates various strategies. These strategies involve managing peak loads in the demand profiles of all consumers by collectively adjusting the operation of equipment, especially those with higher average power consumption, and enhancing power factor using capacitor banks [26]. These techniques result in reduced maximum demand leading to a higher load factor closer to 1 and, consequently, lower current drawn from the supply.

The analysis of the load profiles of the brewery subsections is shown in Fig 11. From the Fig as per data of S4 Table in S1 File, the overall maximum demand for March 2023 is 136.3 kWh from the brewery house plant subsection. Other maximum demands from the remaining subsections were 131.8 kWh, 134.5 kWh, 127.7 kWh, 113.6 kWh, 129.4 kWh, 125.2 kWh, and 114.5 kWh for Malt tower, Beer process cellars, Yeast and Filter cellars, Fermentation and storage vessels, Brew house Auxiliaries, Malt tower and security lights, and Motor control center (MCC) and Brew house office respectively. The diversity factor for the brewery section is 7.4 as computed using equation (2). Furthermore, it can be depicted that these maximum demands occur on weekdays from Monday to Friday. This signifies that during these days all workers are at workplace; therefore, almost all equipment and appliances are in operation resulting in higher energy consumption.

In Fig 12, the analysis of energy consumed by each of the eight different subsections based on weekdays and weekends is presented. It can be observed that the energy usage during the weekdays is higher than 60 kWh from Monday to Friday for all subsections, since, during weekdays apart from the normal production carried out at these subsections, there are additional activities like keg cleaning and spare parts production at the workshop. However, on Saturdays and Sundays, the energy usage is less than 50 kWh as only plant production activities are carried out. Furthermore, most of the office equipment and appliances such as air conditioning, computers, and lighting are off leading to reduced energy usage during weekends. On weekends, Sunday was noticed to utilize the least energy due to few activities performed on this day as shown in Fig 12. Moreover, such an analysis proves highly valuable for identifying the specific times and locations

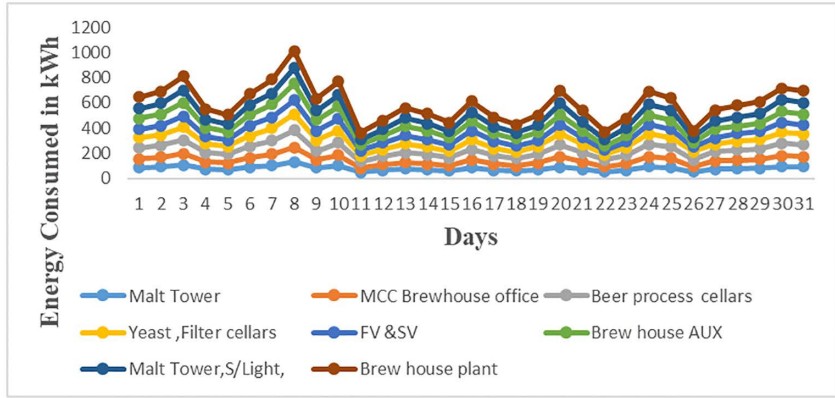

**Fig 11. Brewery subsections' energy consumption for March 2023.**

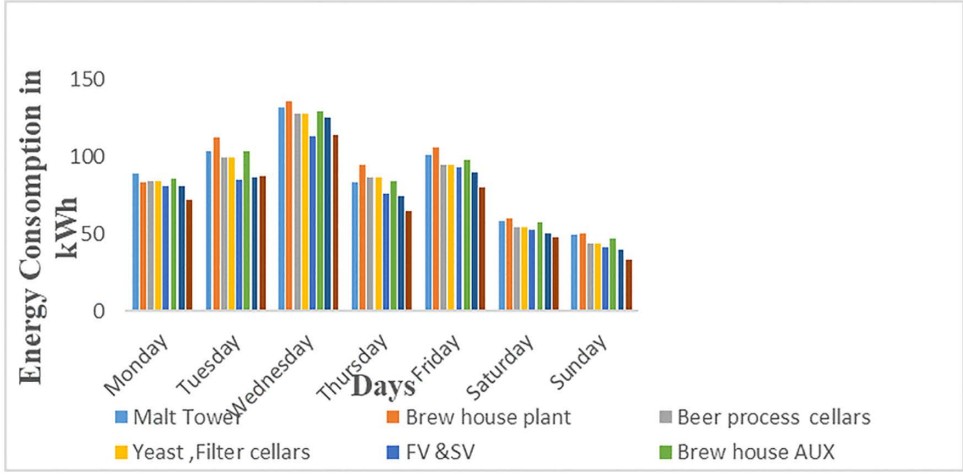

**Fig 12. Energy consumption for the second week of March 2023.**

where the predominant energy consumption occurs. Such data also offers useful insights into the patterns and activities of individuals working within the facilities as per data of S5 Table in S1 File for Fig 12. Therefore, real-time energy demand is crucial as it can aid in optimizing power consumption and pinpointing potential vulnerabilities in power usage within a device, process, or entire facility. Additionally, it significantly lowers operational and maintenance expenses by furnishing essential data such as Peak Load and Base Load for the corresponding facilities.

**2.4.4 Comparison of load parameters without and with the DSEM system.** Industrial equipment and appliances can be categorized according to their energy consumption, operational durations, and end-users' preferences [28]. Based on these three criteria, the connected loads are grouped into schedulable and critical loads, as outlined in Table 3. Furthermore, the schedulable loads are subdivided into time-scheduled equipment loads and power-scheduled equipment loads. The working hours of the brewery section are grouped into three types depending on the production schedules and the activities performed. In production schedules, the working hours are 24 hours. There are two shifts: the first runs from 600 AM to 1800 hours PM, and the second runs from 1800 hours PM to 600 hours AM. For the general activities of the brewery section, the working hours start from 0800 hours AM to 1600 hours PM. The working day load curve for this

**Table 3. The ratings and operation time of different types of loads.**

| Type of loads | Name of Equipment | Power(kW) | Daily Usage (Hrs) |
|---|---|---|---|
| Time-scheduled load equipment | Process water tank pump | 11 | 6 |
| | Caustic pump | 7.5 | 18 |
| | Heat exchange pump | 4 | 24 |
| | CIP return pump | 2.2 | 16 |
| | Caustic dosing pump | 0.18 | 24 |
| | Hydrochloric dosing pump | 0.18 | 20 |
| | Chilled liquor pump | 4.6 | 15 |
| | Caustic pump | 2.2 | 23 |
| | Hops dosing pump | 1.5 | 10 |
| | Recovery water pump | 5.5 | 14 |
| | Floodlights | 3.75 | 21 |
| | Cellars/ filter | 0.75 | 8 |
| | LED tube light | 0.36 | 11 |
| Power-Scheduled load equipment | Lifting sump transfer | 3 | 18 |
| | Fermentation pump | 7.5 | 20 |
| | Hot wort pump | 7.5 | 13 |
| | Cold wort pump | 11 | 15 |
| | Blending pump | 11 | 17 |
| | Glycol pump | 2.2 | 10 |
| | Yeast circulation pump | 4 | 14 |
| | Yeast pitching pump | 4 | 23 |
| | Deaerated pump | 4 | 18 |
| | Motorized screen | 0.37 | 20 |
| | Lifting Sump pump | 3 | 13 |
| | Agitator geared motor | 2.2 | 15 |
| Critical load equipment | Agitator Mash tun motor | 4 | 20 |
| | Circulation whirlpool pump | 4 | 8 |
| | Holding vessel pump | 15 | 20 |
| | Heat exchange pump | 4 | 10 |
| | Beer pump | 11 | 24 |
| | Equalization tank mixer | 0.37 | 20 |

section is illustrated in Fig 13 before the employment of the DSEM. The daily load curve shows a peak demand of 150 kWh at 1100 hours AM. During this time the brewery section is subjected to high losses, system stress, and poor power quality like low power factor. Therefore, the power factor of equipment and appliances is to be automatically improved using IoT. Additionally, schedulable loads are either switched off or programmed to run during off-peak hours to minimize energy consumption. The employment of the DSEM flatten the peak demand as shown in Fig 14.

**2.4.5 Schedulable loads without employment of DSEM.** The categories and parameters of the proposed equipment to be engaged in DSEM are presented in Table 4. Fig 13 shows the daily load curve of equipment without DSEM as data depicted from S6 Table in S1 File. This Figure displays the daily demand for all equipment, featuring three distinct types of load, along with their corresponding operation time slots. This curve is derived from the daily energy consumption plans of each load as per Table 4, incorporating their past equipment usage habits and shift roster patterns. The load curve varies based on weekdays, weekends, and the seasons throughout the year. From the load curve it can be described that

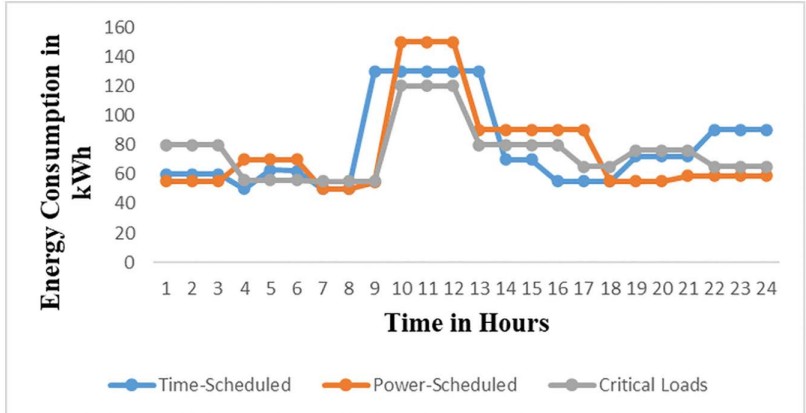

**Fig 13. Daily load curve of equipment without DSEM.**

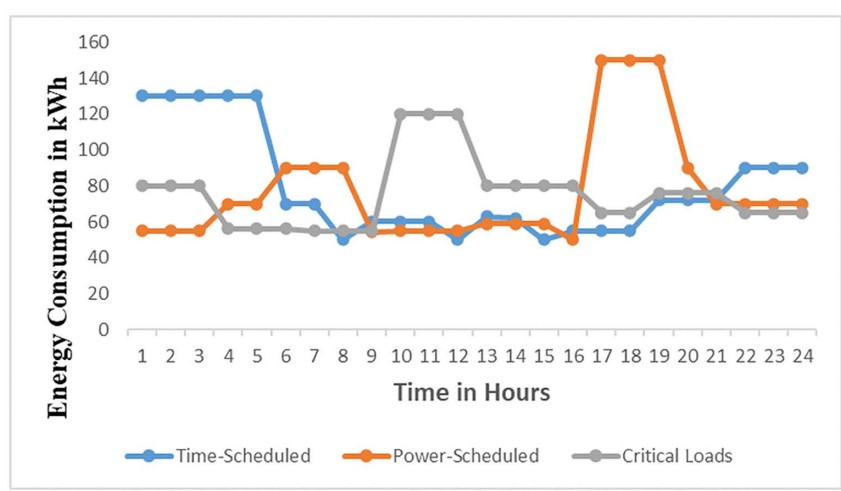

**Fig 14. Daily load curve of equipment with DSEM.**

**Table 4. Daily load operation metrics of solar microgrids during peak operation window (09:30-12:30).**

| Parameter | Unit | Without DSEM | With DSEM | Absolute difference | % change | Primary control mechanism |
|---|---|---|---|---|---|---|
| Peak energy over peak window | kWh | 150 | 120 | 30 | 20.0 | Load scheduling and shedding |
| Average power over peak window | kW | 50.0 | 40.0 | 10.0 | 20.0 | Load scheduling and shedding |
| Load factor | – | 0.47 | 0.58 | 0.11 | 23.4 | Load redistribution |
| Peak-to-average ratio (PAR) | – | 2.143 | 1.714 | 0.429 | 20.0 | Peak flattening |
| Power factor | – | 0.84 | 0.95 | 0.11 | 13.1 | Reactive power compensation |

all types of loads contribute to peak load from 0900 hours AM to 1300 hours PM as the result of unapplied DSEM to shift the schedulable loads to off peak hours. This operation pattern causes stress, reduced system efficiency, and outages in the solar microgrids. Therefore, the suggested DSEM is required to adjust the load curve according to the recommended strategy to flattern the peak demand and align with the generation pattern.

**2.4.6 Schedulable loads with the employment of DSEM.** The goal of scheduling the equipment and appliances is to reduce both peak energy consumption and the average waiting time. Both parts of the equipment and appliances scheduling goal are normalized by adjusting peak to average ratio factor to the minimum value. The aggregate power usage of equipment and appliances must not exceed the capacity of the solar supply microgrids. Therefore, to synchronize demand with supply, it is necessary to minimize the disparity between generation and consumption. Following the implementation of the suggested DSEM, the time slots for shiftable demand have been moved to both before and after noon hours, for maximum solar power utilization, as illustrated in Fig 14 and its data in S7 Table in S1 File. Consequently, the load profile has been enhanced, allowing the customers' shiftable and non-shiftable equipment to be evenly distributed throughout a day resulting in reduction of electrical energy waste.

## 2.5 Mathematical modelling and analysis

**2.5.1 Load profiles and performance metrics.** A load profile can be described as a chronological series of energy consumption values measured at regular intervals over a specified time duration [29]. In the context of energy management and electricity usage, it is a graphical or numerical representation of how the electrical load (the amount of electricity consumed) varies over a specific period. Load profiles provide comprehensive insights into the initial stages of a customer's DSEM initiatives, encompassing aspects such as energy efficiency (EE), peak load control, and demand response, by offering detailed patterns of electrical load consumption [30]. Load profiles can be daily, monthly or yearly depending upon the controlled loads. These load profiles prove valuable in anticipating the patterns of consumption, including peaks, troughs, and trends in energy usage within a specific area, industry, or distribution network. The load profiles describe key terms such as average demand, maximum demand, load factor (LF), diversity factor (DF), and peak to average ratio (PAR) all of which play a crucial role in DSEM. The load factor indicates the proportion of the peak load that is sustained during a specific period. The diversity factor illustrates how the relation between the sum of the individual maximum demands of various subsections to the overall maximum demand of the entire system. Furthermore, the PAR is the ratio of the peak demand to the average demand in a given time [15]. It looks at the concentration of demand during peak periods compared to the average demand, providing information about the load distribution pattern. The load, and diversity factors, are given by equations (1) and (2), respectively [25].

$$LF = \frac{\text{Average Demand}}{\text{Maximum Demand}} \quad (\text{Normally, } LF \leq 1) \tag{1}$$

$$DF = \frac{\text{Sum of Individual Maximum Demand}}{\text{Maximum Demand of the whole system}} \quad (\text{Normally, } DF \geq 1) \tag{2}$$

Moreover, equation (3) can be utilized to model the aggregate PAR for all equipment within a brewery section over the scheduling time horizon [31].

$$PAR = \frac{\text{maximum } (E_i^t)}{\frac{1}{T} \sum_{t=1}^{T} \sum_{i}^{Q} (E_i^t)} \tag{3}$$

Where: $E_i^t$ is the normal energy consumption

T is the time period

The operations and functioning of equipment are rescheduled in a manner that enhances the load factor, peak to average ratio, and consequently lowering the peak demand.

**2.5.2 Modelling of equipment operations.** The connected loads at TBL Mbeya plant can be classified into noncritical loads equipment and critical loads equipment [27]. Noncritical loads are equipment which have essential functions;

do not have an immediate impact on safety, operations, or critical processes if they are switched off during peak load for energy management. They are classified into equipment with time-scheduled characteristics and power-scheduled characteristics based on load control [1]. Examples of non-critical loads equipment include air compressors, utility water heating systems, Clean-in-place (CIP) systems, chillers, and refrigeration systems. Critical loads equipment includes control systems that manage the brewing process, temperature control equipment, pumps and motors, computers and electronics, sensors and instrumentation, centrifuges, and filtration systems. For the operation of equipment the period for a day is H = [1, 2, 3, ..., T], with an overall period of 24 hours. Where 1 is the first hour and T is the end 24$^{th}$ hour. The demand has three categories of equipment EQ = [$EQ_s^T \cup EQ_s^P \cup EQ_{CR}^{PF}$], where [$EQ_s^T, EQ_s^P, EQ_{CR}^{PF}$], indicate time-scheduled load, power-scheduled load, and critical load equipment, respectively. In equipment (EQ) the starting time is ($A_{EQ}$) and the finishing operation time is ($B_{EQ}$). Equipment on/off position is shown by ($X_{eq}^t$), the rest operation hours ($z_{eq}^t$), and waiting hours ($\omega_{eq}^t$). Moreover, the energy consumption is ($E_{eq}^{EQ}$) and the cost per unit is ($\rho_t^{CR}$). For t < $A_{EQ}$ and t > $B_{EQ}$ time is zero because the energy consumption outside the operation period is zero. The mathematical models are performed considering the types of load.

**2.5.3 Time-scheduled equipment.** These equipment exhibit flexible operational schedules, allowing for operation delays or advances within the scheduled period, and operate with a predetermined power rating ($P_{eq}^r$) for specified operating hours ($H_{eq}^t$). The functioning of such equipment can be postponed, rescheduled, or stopped as needed. The mathematical models representing the current and next status of equipment with time-scheduled capabilities are as per equations (4) and (5) respectively. Equations (6) and (7) are utilised to calculate the energy consumption and utility bill areas for time-scheduled equipment [1,31].

$$X_{eq}^t = (H_{eq}^t, A_{eq} - B_{eq} - H_{eq}^t + 1) \tag{4}$$

$$X_{eq}^{t+1} = \begin{cases} (z_{eq}^t, \omega_{eq}^t - 1) & \text{if } X_{eq}^t = 0, \ \omega_{eq}^t \geq 1 \\ (z_{eq}^t - 1, \omega_{eq}^t) & \text{if } X_{eq}^t = 1, \ z_{eq}^t \geq 1 \end{cases} \tag{5}$$

$$E_{eq}^{EQ} = \sum_{eq \in EQ_s^T} \sum_{t=1}^{T} (P_{eq}^r \times X_{eq}^t) \tag{6}$$

$$C_{eq}^{EQ} = \sum_{eq \in EQ_s^T} \sum_{t=1}^{T} (P_{eq}^r \times X_{eq}^t \times \rho_t^{CR}) \tag{7}$$

Where: $X_{eq}^t$ is the equipment on/off position

$A_{EQ}$ is the starting time

$H_{eq}^t$ is the operating hours

$B_{eq}$ is the finishing operation time

$z_{eq}^t$ is the rest operation hours

$\omega_{eq}^t$ is the waiting hours

$E_{eq}^{EQ}$ is the energy consumption

$EQ_s^T$ is the time scheduled load equipment

$P_{eq}^r$ is the power rating of equipment

$C_{eq}^{EQ}$ Total cost of energy consumed by equipment

$\rho_t^{CR}$ the cost of energy consumed per unit

**2.5.4 Power- scheduled equipment.** Power-scheduled equipment function with adaptable power levels within the designated scheduling period and do not operate beyond the specified scheduling time horizon. They operate with power levels ranging from $P_{min}$ to $P_{max}$, representing the minimum and maximum power ratings, respectively. The current and subsequent states of power-scheduled equipment are determined by using Equations (8) and (9), respectively. The net energy consumption and utility bill for power-scheduled equipment are calculated using Equations (10) and (11), respectively [1,31].

$$X_{eq}^t = (H_{eq}^t, A_{eq} - B_{eq} - H_{eq}^t + 1) \tag{8}$$

$$X_{eq}^{t+1} = \begin{cases} (z_{eq}^t - 1, 0) & \text{if } X_{eq}^t = 1, \ z_{eq}^t \geq 1 \\ \left( P_{eq}^{r\ min} \leq P_{eq}^r \leq P_{eq}^{r\ max} \right) & \text{if } X_{eq}^t = 1, \ z_{eq}^t \geq 1 \end{cases} \tag{9}$$

$$E_{eq}^{EQ} = \sum_{eq \in EQ_s^P} \sum_{t=1}^{T} (P_{eq}^r \times X_{eq}^t) \tag{10}$$

$$C_{eq}^{EQ} = \sum_{eq \in EQ_s^P} \sum_{t=1}^{T} (P_{eq}^r \times X_{eq}^t \times \rho_t^{CR}) \tag{11}$$

Where: $P_{eq}^{r\ min}$ and $P_{eq}^{r\ max}$ is the minimum and maximum power rating of equipment respectively

$EQ_s^P$ is the power scheduled load equipment

**2.5.5 Critical load equipment.** Critical load equipment refers to devices or systems that are essential for the proper functioning of a facility or operation and whose failure can have significant consequences on safety, operations, or the overall operation [32]. These are the components that, if disrupted, could lead to serious consequences, including safety hazards, operational downtime, or financial losses. The net energy consumption and utility bill of the critical load equipment are calculated from equations (12) and (13), respectively [1,31].

$$E_{eq}^{EQ} = \sum_{eq \in EQ_{CR}^P} \sum_{t=1}^{T} (P_{eq}^r \times X_{eq}^t) \tag{12}$$

$$C_{eq}^{EQ} = \sum_{eq \in EQ_{CR}^P} \sum_{t=1}^{T} (P_{eq}^r \times X_{eq}^t \times \rho_t^{CR}) \tag{13}$$

Where: $EQ_{CR}^P$ is the critical load equipment

**2.5.6 Power factor correction analysis.** The power factor (PF) represents the relationship between the actual power (kW) and the total apparent power (kVA) consumed by an electrical load [15]. Inadequate control of power factor results in numerous problems, such as larger kVA equipment ratings, increased driver size, challenges in low voltage control, elevated copper loss, and reduced power handling capacity. Furthermore, it leads to an imbalance between the power supply and demand sides. Mathematically is indicated by equation (14).

$$\cos\varnothing = \frac{P}{S} = PF \tag{14}$$

Where: $\varnothing$ is the power factor angle measured in degrees, determined by the phase difference between the voltage and current signals.

P is real power in Watts (W), S is apparent power in Volt-amperes (VA). The apparent power is the algebraic sum of real and reactive powers as given by equation (15)

$$S = P + jQ \tag{15}$$

Q is the reactive power

The calculation of the reactive power required to achieve the target power factor is as per equation (16) [33]

$$Q = P \times (\tan\left(\varnothing_{new}\right) - \tan\left(\varnothing_{old}\right)) \tag{16}$$

## 2.6 Validation and comparative analysis

**2.6.1 Error analysis and discussion.** Sensor calibration establishes the reliability of measured electrical parameters before performance assessment. The PZEM-004T energy meter was evaluated against an MS2203 digital clamp meter under identical operating conditions. Both instruments were connected to the same circuit and supplied from a stable source. Measurements covered voltage, current, active power, power factor, apparent power, and energy. This arrangement isolates sensor error from load variability and wiring effects. TheMS2203serves as the reference instrument due to its higher measurement resolution. Readings from the PZEM-004T were recorded simultaneously and compared parameter by parameter. Measurement error is expressed as a percentage difference relative to the reference instrument as per equation (17) [34].

$$\%Error = \frac{|S_{PZEM-004T} - S_{MS2203}|}{S_{MS2203}} \tag{17}$$

Where: $S_{MS2203}$ is calculated based on voltage and current of the standard digital clamp meter, $S_{PZEM-004T}$ is obtain based on measurement using PZEM-004T sensor. This formulation allows direct evaluation of sensor bias under practical load conditions. Uncertainty in estimated energy savings is quantified using confidence intervals to bound expected performance. The interval is computed as shown in equation (18)

$$CI = \overline{X} \pm Z \times \left(\frac{\sigma}{\sqrt{n}}\right) \tag{18}$$

Where: CI is the confidence interval;
$\overline{X}$ is the mean estimated energy savings; Z is the Z-score (for a given level);
$\sigma$ is the standard deviation of savings measurements; n is the number of observations.
For a specified error margin, the expected savings range is as per equation (19)

$$CI = \overline{X} \pm (EM \times \overline{X}) \tag{19}$$

Where: EM is the error margin.

Data collection over the two-month testing period supports assessment of measurement stability and control consistency. Cross-verification with independent instruments limits systematic bias, while repeated observation under comparable operating conditions supports repeatability. Load balancing effects are evaluated through observed reductions in reactive power demand and stabilised current profiles. These outcomes indicate improved operational reliability under sustained operation rather than isolated test conditions.

**2.6.2 Comparison of load parameters without and with the DSEM system.** This section evaluates changes in load performance before and after applying the demand-side energy management system. The analysis focuses on cumulative energy over the defined peak operating window from 09:30–12:30 and on standard load indicators. Without the DSEM system, cumulative energy consumption during the peak window reached 150 kWh, as summarized in Table 4. This level reflects simultaneous operation of critical and non-critical loads during peak production hours. After applying DSEM, peak-window energy decreased to 120 kWh, as reported in Table 4. This change represents a 20% reduction in cumulative energy during peak operation. Direct control of non-critical loads drives this reduction. Production schedules and critical processes remain unchanged. The reduction magnitude aligns with reported reductions of 21.57% in comparable industrial applications [27]. The load factor increased from 0.47 to 0.58, which corresponds to a 23.4% improvement, as compared to the LF increased from 0.536 to 0.545, an increase of 1.6% by [35]. Our result is higher probably due to the environmental conditions such as temperature and humidity during measurement [36]. This change indicates improved alignment between average demand and peak demand across the operating day. The peak-to-average ratio decreased from 2.143 to 1.714, which represents a 20% reduction. This result indicates reduced demand concentration during the peak window and confirms effective peak flattening, consistent with trends reported in [28]. Our result is lower possibly due to the ratings of various loads and their power consumption [15]. Table 4 summarizes these energy-related indicators and shows coherent improvement under DSEM control.

Power quality indicators show a distinct response to the applied control actions. The power factor increased from 0.84 to 0.95 after automatic reactive power compensation, as shown in Table 4. This change corresponds to a 13.1% improvement and aligns with power factor values reported in related demand-side management studies [37]. The improvement reduced current drawn from the supply and lowered apparent power demand. It also increased available inverter capacity during peak operation. Cumulative energy consumption over the peak window remained governed by load scheduling rather than reactive power control. The results therefore separate energy reduction effects from power quality effects. Load shedding drives reductions in real energy use during peak periods. Power factor correction drives reductions in apparent power and current. Table 4 consolidates these outcomes and clarifies the distinct roles of the two control actions within the DSEM framework.

# 3 Results and discussion

## 3.1 Simulation results

### 3.1.1 Power factor correction and load shedding.
The simulation of the proposed system was carried out in Proteus 8.15 incorporating IoT for remote control of equipment and loads. The proposed system enables the equipment and power consumption at the end users to be remotely controlled by the supply of solar microgrids. Solar microgrids supply can view customers' electrical equipment operation and meter readings in real time from a web-application. The meter readings in real time at the end users are also displayed at the supply solar microgrids where the remote monitoring and control of equipment and appliances are performed. The remote monitoring at the end users is based on power factor correction and monitoring of noncritical loads. The assumptions made during simulations are; the accuracy of the current and voltage sensors is according to the manufacturer's specifications, the surrounding temperature and humidity parameters are constant.

When the overall power factor of customers is less than 0.95, the proposed system send an email to alert the customers on low power factor. If the customer does not respond, the supply solar microgrids will activate the capacitor bank to normalize the power factor. The switching on of capacitor bank reduces current drawn from the solar supply source from

4.93 A to 2.87 A, the reduction of 41.78%. Consequently, apparent power is minimized from 1143.38 VA to 667.80 VA a decrease of 41.59%, and the real power is reduced from 272 W to 265 W a reduction of 2.57% as illustrated in Figs 15 and 16. A reduction of 41.59% is observed in apparent power demand following reactive power compensation. The corresponding reduction in real power remains limited, confirming that the primary benefit arises from reactive power mitigation rather than direct energy reduction. Furthermore, the power factor is improved from 0.83 to 0.95 as shown on the display of Figs 15 and 16 an enhancement of 0.12 (12.63%). The reduction for both current and apparent power is almost the same during the process with slight variation in voltage. Under this condition, the power quality of the system is improved and the efficiency of energy consumption is improved leading to low power losses and system stress is reduced. Furthermore, the decrease in current signifies that the system operates at high efficiency, thus minimizes power imbalances between the supply solar microgrids and end users.

Energy consumption at the customers can be controlled by switching off noncritical loads during peak periods without affecting production and customer comfort. The isolation of these loads is necessary for further reduction of energy consumption as depicted in Fig 17. From the Figure, it can be observed that apparent power consumed is decreased from 667.80 VA to 497.94 VA, a reduction of 25.44%. Moreover, the current drawn from the supply is minimized from 2.87 A to 2.14 A, a decrease of 25.35%, similarly, the real power is minimized from 265 W to 212 W, a reduction of 20%. However, these results are subject to sensor accuracy (±5%) and environmental conditions that may impact real-world implementation [36]. These noncritical loads are scheduled to operate during off peak hours, thus flattening the load curve during peak hours. Shifting these loads from peak hours to off peak hours reduces power outages and postpone the construction of new power generation of the supply solar microgrids.

### 3.2 Experimental results

**3.2.1 Hardware prototype performance.** If a customer's total power factor falls below 0.95 for instance, dropping to 0.84 as shown in Fig 18 of the prototype. The proposed system sends an email notification to alert customers about

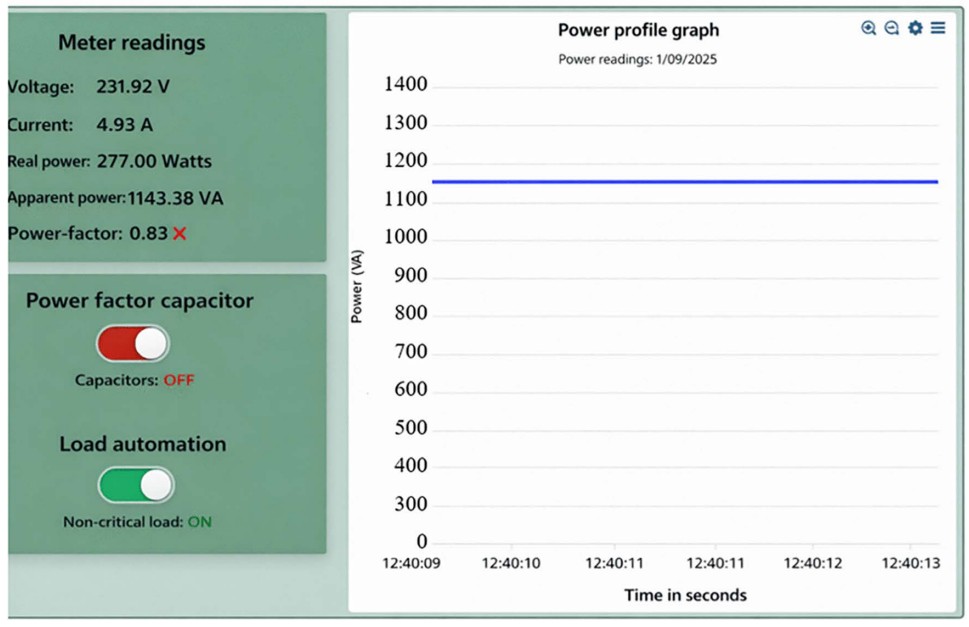

**Fig 15. High power consumption when the capacitor is off.**

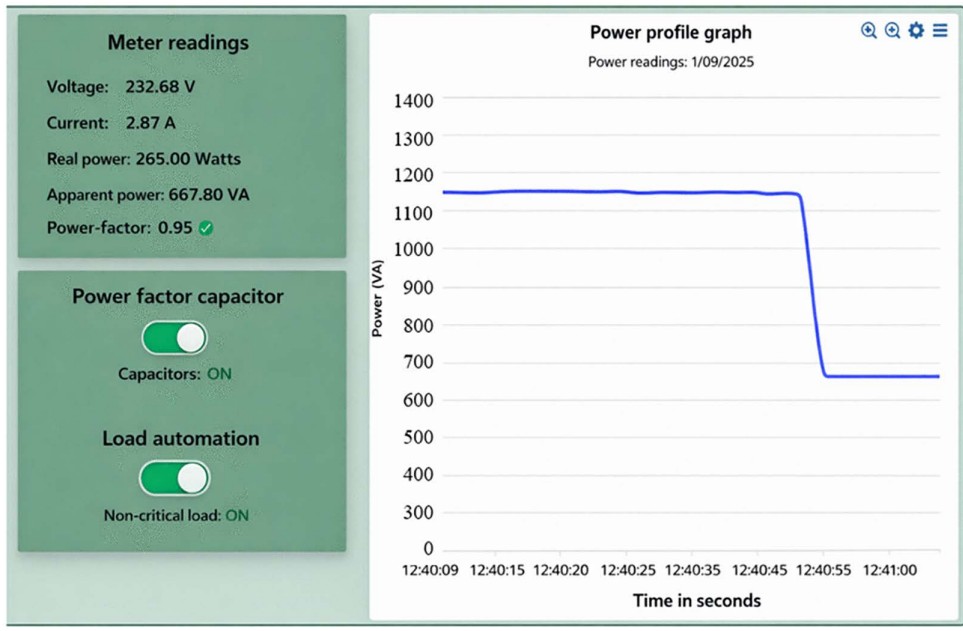

**Fig 16. Power reduction when capacitor banks are switched on.**

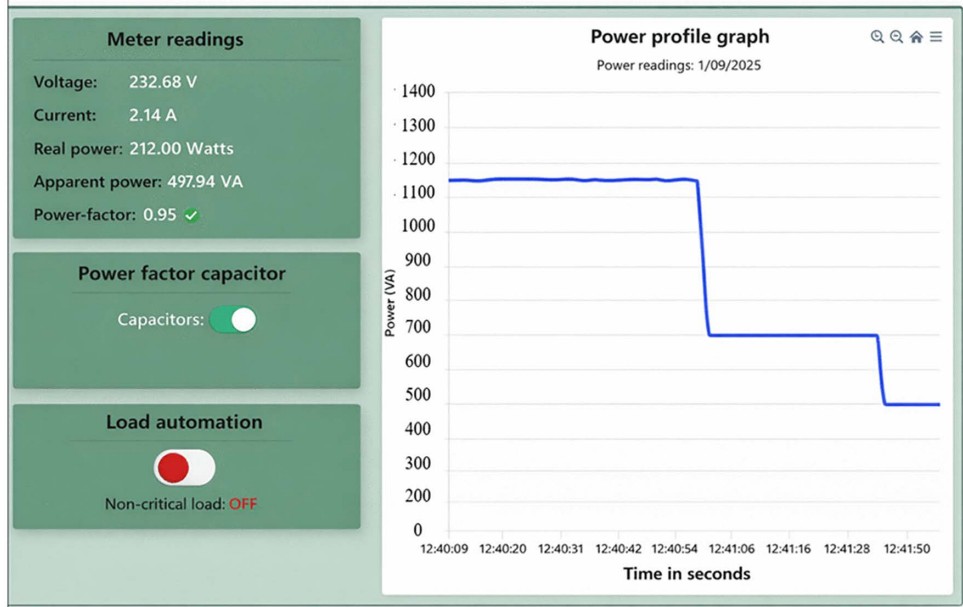

**Fig 17. Further power reduction by switching off noncritical loads.**

the issue. If the customer does not take action, the solar microgrid system automatically activates the capacitor bank to stabilize the power factor as illustrated in Fig 19. Activating the capacitor bank decreases the current drawn from the solar supply source from 4.99 A to 2.94 A, a reduction of 41.08%. Consequently, the apparent power reduces from 1159.18 VA

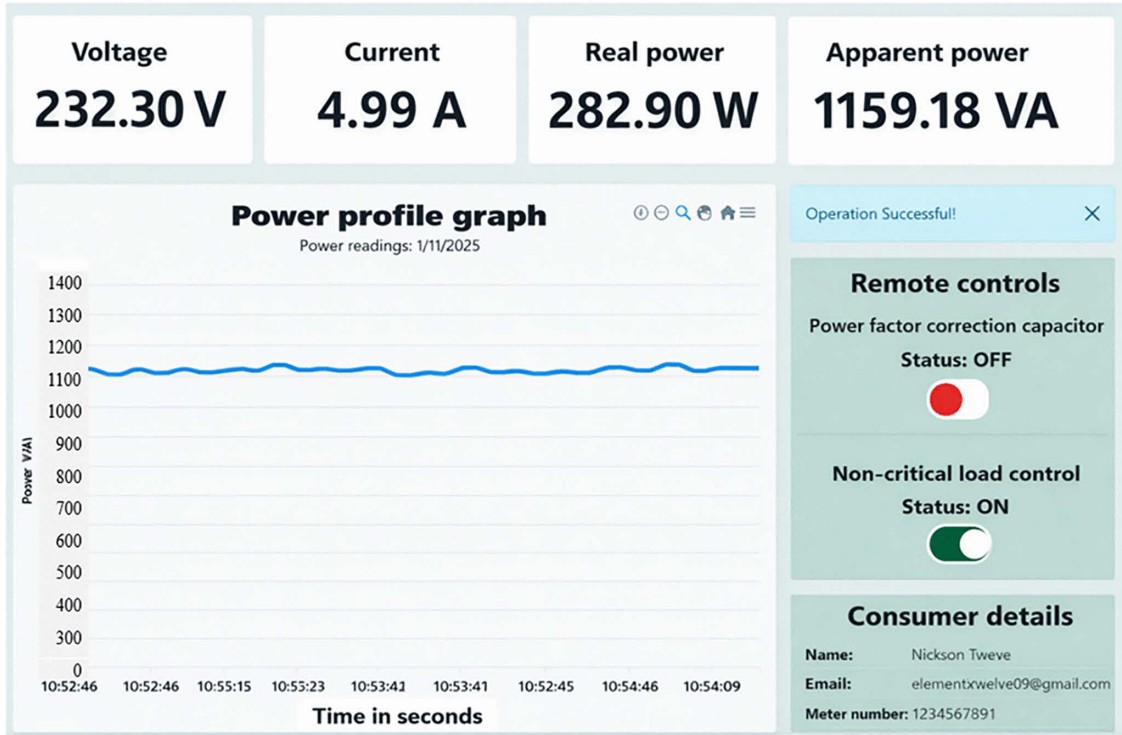

**Fig 18. High power consumption when the capacitor is off.**

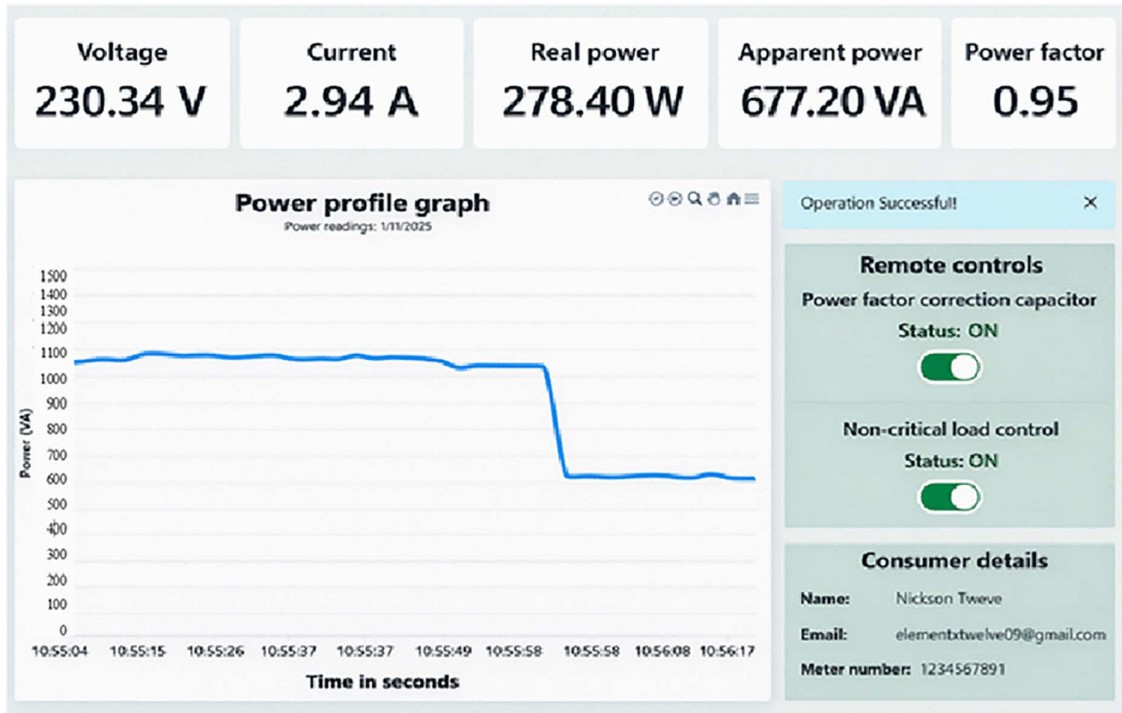

**Fig 19. Reduced power consumption when the capacitor is switched on.**

to 677.20 VA, reflecting a 41.58% decrease. Also, the real power was reduced from 282.90 W to 278.40 W, a reduction of 1.6%. Moreover, the power factor is significantly improved from 0.84 to 0.95, representing an enhancement of 0.11 (13.10%), as shown in Figs 19 and 20. The decrease in apparent power is higher than that of real power because the apparent power is the algebraic sum of real power and reactive power as per equation (15), and the more significant reduction is due to the decrease in reactive power. The reactive power is alleviated by power factor enhancement. This reduction in both current and apparent power minimizes energy waste, thereby conserving energy during peak demand periods.

Energy consumption by customers can be optimized by turning off noncritical loads during peak periods without affecting production or customer comfort. Isolating these loads is important for achieving additional energy savings, as demonstrated in Fig 20. The Figure shows that the current drawn from the supply drops from 2.94 A to 2.72 A, a 7.48% reduction. Similarly, a decrease in apparent power consumption from 677.20 VA to 632.40 VA, representing a 6.62% reduction. Furthermore, the real power is reduced from 278.40 W to 213.10 W a decrease of 23.46%, aligning with previous studies on IoT-based power optimization [27] which reduced power by 21.57%. Noncritical loads are rescheduled to operate during off-peak hours, effectively flattening the load curve during peak times. This load shifting reduces power outages and delays the need for additional power generation capacity in supplying solar microgrids.

**3.2.2 Comparison between simulation and experimental results.** Simulation results are compared with experimental measurements to assess consistency in observed trends rather than numerical equivalence. Minor deviations are attributed to sensor tolerance, environmental variation, and hardware non-idealities. When all loads are switched on at the end user, the power consumption is at its peak; therefore, demand-side energy management through power quality improvement and load shedding is required to reduce power consumed. From Table 5 it can be seen that as the power quality is enhanced through automatic power factor control by switching on the capacitor banks when it is low. Under simulation, the power factor increases from 0.83 to 0.95, while in the experimental setup, it increases from 0.84 to 0.95. This improvement leads to a decrease in real power consumption, apparent power, and current drawn from the solar supply, with the reduction in each case detailed in Table 5. Improving the power factor not only reduces the amount

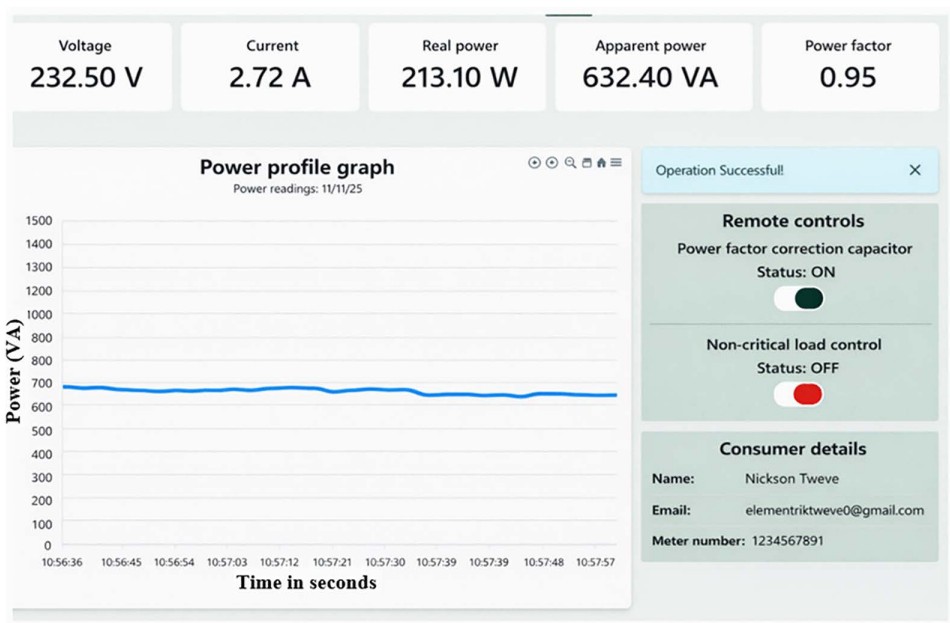

**Fig 20. Reduction of power by load shedding.**

**Table 5. Comparison of simulation and experimental parameters.**

| S/N | Cases | Implementation | Voltage (V) | Current (A) | Real power (W) | Apparent power (VA) | Power factor |
|---|---|---|---|---|---|---|---|
| 1. | Capacitor is off when all loads are on. | Simulation | 231.92 | 4.93 | 272 | 1143.38 | 0.83 |
| | | Experimental | 232.3 | 4.99 | 282.9 | 1159.18 | 0.84 |
| 2. | Capacitor is on when all loads are on. | Simulation | 232.68 | 2.87 | 265 | 667.8 | 0.95 |
| | | Experimental | 230.34 | 2.94 | 278.4 | 677.2 | 0.95 |
| 3. | Capacitor is on when non critical loads are off. | Simulation | 232.68 | 2.14 | 212 | 497.94 | 0.95 |
| | | Experimental | 232.5 | 2.72 | 213.10 | 632.4 | 0.95 |

of power consumed but also minimizes system losses, which can cause overheating and potential damage to equipment and appliances due to excessive current drawn from the supply. Moreover, the automatic load shedding of noncritical loads decreases power consumption during peak loads. The decrease in power consumption resulting from switching off noncritical loads is illustrated in the Table. For both simulation and experimental studies, a decrease was observed in current, real power, and apparent power, while the power factor remained constant. The power factor remained unchanged since the capacitor was used for power factor correction. Additionally, critical loads categorized as resistive, signifying there is no phase displacement between current and voltage. As a result, the capacitor remained unaffected when switched off, as indicated in Table 5.

### 3.3 Comparative analysis

**3.3.1 Comparison with traditional demand response models.** The IoT-based DSEM system was evaluated against conventional price-based demand response models that rely on voluntary participation. The proposed system employs deterministic control rules rather than mathematical optimization algorithms. Performance improvements are achieved through automated decision execution rather than optimal scheduling. While traditional DR approaches have shown some success in reducing peak demand, their effectiveness is often limited by delayed user responses and dependence on voluntary engagement [24]. Studies indicate that price-based DR programs achieve only a 20% reduction in peak demand, whereas the proposed IoT-based system reduced apparent power by 41.58%, demonstrating a 21.58% improvement in efficiency [1,38]. However, the case study was not the same and the percentage reduction was different. The results indicated a similar trend in energy reduction when DSEM is employed.

Consumer compliance in traditional DR models remains inconsistent, as many users opt out despite financial incentives [39]. The IoT-based DSEM system eliminates this issue through real-time automated intervention, ensuring dynamic enforcement of power factor correction and load management. Removing reliance on voluntary user decisions guarantees sustained energy efficiency improvements and grid stability. Unlike traditional systems that passively respond to price signals, the IoT-based approach actively processes real-time sensor data, adjusting power consumption to prevent sudden load surges and optimize energy use without human intervention.

Two-way communication between consumers and the microgrid further differentiates the IoT-based system from traditional DR models. Conventional systems operate on fixed schedules and price incentives, whereas the proposed system dynamically adjusts energy demand, facilitating real-time load shedding and capacitor switching based on immediate grid conditions. Research has demonstrated that real-time demand-side management increases grid stability by up to 30%, supporting the observed improvements in this study [40]. The comparative performance indicators in Table 5 highlight the superiority of the IoT-based DSEM system, demonstrating higher efficiency in power factor correction, peak demand reduction, and overall system adaptability. The ability to automatically optimize load distribution significantly enhances operational efficiency, reinforcing the shift from voluntary price-based models to automated, data-driven demand-side energy management solutions.

**3.3.2 Evaluation against literature.** Results were evaluated against existing IoT-based DSEM research, focusing on efficiency gains, system scalability, and novel contributions. Prior studies on IoT-driven DR systems have largely focused on residential energy management, with limited applicability to industrial microgrid environments [41]. The proposed system extends IoT-based energy management to industrial settings, demonstrating that automated load balancing and real-time power factor correction are feasible on a larger scale. The improved scalability and industrial validation of the proposed approach address a key research gap, as most existing models struggle with high-load industrial applications due to network latency and processing limitations [42].

The power factor correction capabilities of the proposed system align with research conducted by [43,44], which reported that demand-side optimization techniques increased load factor by 10% and reduced the PAR by 12%. The proposed system achieved a 23.4% load factor increase, while delivering a higher reduction in apparent power (41.58%), surpassing previous benchmarks. This suggests that real-time capacitor switching combined with IoT-enabled load control yields more significant efficiency gains than static DR techniques. Previous research has relied on time-scheduled power adjustments, which lack the flexibility to respond to real-time fluctuations in energy demand [38]. The proposed system integrates real-time sensor feedback, allowing instantaneous corrective actions that enhance power factor and reduce energy losses more effectively than traditional time-based DR systems.

The scalability of the IoT-based system was a key evaluation criterion. Traditional IoT-based DR frameworks frequently encounter hardware limitations and unreliable network performance, making them unsuitable for large-scale deployment [45]. The proposed system successfully leveraged low-latency MQTT-based communication protocols, and cloud integration, enabling seamless real-time energy monitoring and control across multiple industrial loads. Adaptability to different industrial environments is enhanced, overcoming constraints typical of localized residential DR systems. The scalability potential of the system is reinforced in Table 6 compares previous models with our system, showing better response times, adaptability, and energy efficiency. Scalability is evaluated qualitatively based on system architecture. The control logic processes each monitored load independently using fixed threshold rules, resulting in constant decision time per node. This structure supports incremental expansion through additional IoT nodes without increasing control complexity at individual endpoints. Remote monitoring, analysis, and control of energy consumption via a cloud-based interface ensure long-term efficiency improvements and adaptability to evolving energy demand patterns [46].

Cloud integration improved system accessibility but introduced communication latency during remote operation. Local threshold-based control continued to execute at the controller level and remained unaffected by these delays. Communication latency influenced remote monitoring and command transmission. This observation supports future integration of edge-level processing to reduce dependence on cloud communication for time-sensitive control tasks, reducing dependency on real-time internet connectivity [47]. Strengthening cybersecurity measures through blockchain-based authentication could further enhance data integrity and prevent unauthorized access, ensuring the reliability of IoT-driven energy management platforms [48].

The findings in Table 6 illustrate the significant advancements provided by the IoT-based DSEM system over existing methods basing on the listed key indicators. The combination of real-time power factor correction, automated load control,

Table 6. Comparative analysis of IoT-based DSEM system and traditional demand response models.

| Performance Indicator | Traditional Demand Response (DR) Models | Proposed IoT-Based DSEM System | Improvement (%) | Reference |
|---|---|---|---|---|
| Peak Demand Reduction | 20% (Price-Based DR) | 41.58% | +21.58% | [24,49] |
| Power Factor Improvement | 12% (from 0.87 to 0.99) | 13.1% (from 0.84 to 0.95) | +1.1% | [37] |
| Load Factor Increase | 5–10% | 23.4% | +13.4% | [50] |
| Peak-to-Average Ratio (PAR) Reduction | 12% | 20% | +8% | [43] |

and cloud-based monitoring presents a highly effective energy management solution for industrial microgrids, demonstrating a substantial improvement in demand-side optimization strategies. The real-time intervention mechanism, absent in many previous studies, contributes to greater energy efficiency and operational stability, reinforcing the case for broader adoption of IoT-driven energy management solutions in industrial applications.

### 3.4 Discussion of findings

**3.4.1 Implications for industrial applications.** Demand Side Energy Management (DSEM) involves strategies and technologies to optimize energy use, cut costs, and improve efficiency especially critical in energy-intensive industries like manufacturing, steel, chemical processing, and data centers, where energy expenses significantly impact operations [51]. High-energy industries benefit from DSEM by reducing peak demand charges through load shifting and demand response, optimizing energy use by identifying inefficiencies, and using real-time monitoring to adjust operations. DSEM also allows shifting non-essential loads to off-peak hours, lowering energy costs and improving efficiency [6]. Participating in DR programs allows industries to reduce consumption during peak demand in exchange for financial incentives. DSEM offers significant benefits in high-energy industries, including cost savings, improved operational efficiency, and sustainability. By using real-time monitoring, demand response, predictive maintenance, and renewable energy, industries can enhance energy management and comply with regulation [52].

The DSEM system, integrated with Automatic Power Factor Correction (APFC), optimizes energy use, improves grid stability, and enhances operational efficiency. APFC reduces reactive power demand, improving power utilization and reducing strain on the grid. Real-time monitoring with sensors and microcontrollers detects power factor variations and activates capacitor banks or inductive loads to correct them [37]. Modern DSEM systems use IoT and AI analytics for real-time monitoring, automated alerts, integration with smart grids, and predictive energy demand forecasting [15]. Scalability in DSEM is essential for industries with fluctuating energy needs, large operations, and future growth. It ensures cost savings, efficiency, and sustainability, particularly in industries with high cooling and power demands or multi-location operations.

**3.4.2 Benefits of mandatory and incentive-based participation.** Mandatory demand response improves grid reliability and effectiveness by ensuring a consistent baseline of demand reduction during peak periods, enabling better grid management and planning [53]. Mandatory demand response enhances grid stability by providing predictable and consistent demand reduction, enabling better forecasting and planning. It ensures broad participation across residential, commercial, and industrial sectors, strengthening grid resilience and supporting systems like solar microgrids during high demand or disruptions. This approach also prevents free-rider issues, promoting fairness and distributing operational and financial benefits more evenly. Consistent demand response helps flatten peak demand, reducing reliance on costly peaking power plants and lowering system costs. Predictable load reductions also encourage investment in smart grid technologies and energy storage, boosting overall efficiency [16]. However, mandatory participation may reduce comfort and preferences of customers [54]. Privacy concerns or distrust might create resistance, especially if participation feels forced, therefore, privacy and data use transparency are essential for adoption. Incentives might not be accessible or appealing for all customers, especially those with fixed needs or lower incomes [21]. Thus, mandatory participation in demand response programs creates a more predictable, equitable, and effective framework for managing energy demand. It not only stabilizes the grid and supports the integration of renewable resources but also promotes economic efficiency and fair distribution of the demand reduction burden.

## 4 Conclusions

This study investigated an IoT-based demand-side energy management framework tailored to industrial solar microgrids. The work focused on electrical performance rather than aggregated energy metrics, with explicit separation of active power, reactive power, and apparent power. The framework integrated real-time power factor monitoring, automated

reactive power compensation, and prioritized load control within a unified architecture. Simulation and prototype results confirmed that reactive power mitigation significantly reduced apparent power demand and current flow, releasing inverter capacity and reducing system stress. Real power reduction was achieved through automated shedding of non-critical loads during peak periods, producing measurable energy savings without disrupting industrial operations. Analysis of historical and experimental data showed consistent improvements in load factor and peak-to-average ratio, indicating a flatter demand profile and improved utilization of solar microgrid resources.

The control strategy employed deterministic threshold-based rules with low computational overhead. Stability was verified empirically through sustained operation without oscillatory behavior. The system architecture supports incremental scaling at facility level through distributed IoT nodes, although formal scalability analysis and closed-loop stability proofs were outside the scope of this study. The findings confirm the technical feasibility of automated industrial demand-side energy management based on real-time electrical parameters rather than voluntary consumer response. The framework addresses key limitations of price-based and incentive-based demand response in industrial environments, where production continuity constrains behavioral flexibility. Future work will extend the architecture through optimization-based scheduling, machine-learning-assisted load disaggregation, and edge-level control to improve scalability, adaptability, and predictive demand management. Moreover, additional experiments for different load profiles are recommended for future research.

## Supporting information

**S1 File. Supporting Information file contains S1 Fig, S2-S7 Table.**
(DOCX)

## Acknowledgments

The authors gratefully acknowledge the support of Mbeya University of Science and Technology and the Ministry of Education, Culture, Sports, Science and Technology, provided through the Higher Education Economic Transformation Project. Moreover, the authors sincerely appreciate the AI tools as ChatGPT was used for checking grammar in the whole manuscript.

## Author contributions

**Conceptualization:** Exaud Tweve, Godiana Philipo.

**Data curation:** Exaud Tweve.

**Formal analysis:** Exaud Tweve, Godiana Philipo.

**Investigation:** Thomas Kivevele, Baraka Kichonge, Godiana Philipo.

**Methodology:** Exaud Tweve, Baraka Kichonge, Godiana Philipo.

**Resources:** Thomas Kivevele.

**Software:** Exaud Tweve.

**Supervision:** Thomas Kivevele, Baraka Kichonge.

**Validation:** Thomas Kivevele, Baraka Kichonge.

**Visualization:** Thomas Kivevele.

**Writing – original draft:** Exaud Tweve.

**Writing – review & editing:** Thomas Kivevele, Baraka Kichonge, Godiana Philipo.

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
