## [Decision Letter · Decision Letter 0]

3 Feb 2026

Dear Dr. Kivevele,

Thank you for submitting your manuscript to PLOS ONE. After careful consideration, we feel that it has merit but does not fully meet PLOS ONE’s publication criteria as it currently stands. Therefore, we invite you to submit a revised version of the manuscript that addresses the points raised during the review process.

We look forward to receiving your revised manuscript.

Kind regards,

Zhengmao Li

Academic Editor

PLOS One

**Journal Requirements:**

3. We note that your Data Availability Statement is currently as follows:

“All relevant data are within the manuscript and its Supporting Information files.”

5. Please amend either the title on the online submission form (via Edit Submission) or the title in the manuscript so that they are identical.

6. Please include a caption for figures 2, 3, 4, 5, 6, 7, 8, 9, 10, 11, 17, and 19.

**Additional Editor Comments:**

Please revise

Reviewers' comments:

Reviewer's Responses to Questions

**Comments to the Author**

1. Is the manuscript technically sound, and do the data support the conclusions?

Reviewer #1: Yes

Reviewer #2: Partly

2. Has the statistical analysis been performed appropriately and rigorously?

Reviewer #1: Yes

Reviewer #2: Yes

3. Have the authors made all data underlying the findings in their manuscript fully available?

Reviewer #1: Yes

Reviewer #2: Yes

4. Is the manuscript presented in an intelligible fashion and written in standard English?

Reviewer #1: Yes

Reviewer #2: Yes

Reviewer #1: This paper presents an IoT-based demand-side energy management system for industrial solar microgrids. It combines power factor correction and non-critical load control, and it is supported by both simulation and a hardware prototype. The strongest evidence is the experimental trend that PF correction mainly reduces apparent power and current, while load shedding produces the larger real power reduction, but several results need clearer definitions and consistent units to avoid confusion.

1. The separation of real, reactive, and apparent power is a good framing choice, and Eq. (15) helps explain why apparent power falls more than real power. The manuscript sometimes shifts between “peak demand reduction” and “apparent power reduction” without stating which metric is being discussed in that paragraph, especially around the reported 41.58% reduction.

2. The paper benefits from having both Proteus simulation and a physical prototype, and the ±1.5% calibration statement is helpful. It would be stronger if Section 2.3.1 included one small table or example readings showing MS2203 vs PZEM-004T agreement for voltage, current, and PF, not only the final accuracy claim.

3. The paper states “peak load is 150 kWh” during 09:30–12:30, which looks like an energy unit used as power, so please clarify whether this is peak hourly energy, peak power, or another definition, and keep it consistent with Table 3.

4. One discussion that could strengthen the paper is to comment on whether their IoT DSEM framework could incorporate an explicit “risk limit” instead of hard thresholds, similar in spirit to CVaR-based safety constraints in robust constrained MDP formulations (see a robust safe reinforcement learning-based operation method for hybrid electric-hydrogen energy system risk-based dispatch considering dynamic efficiency characteristics of electrolysers). Also, since your controller already emphasizes stability without formal proofs, a short comparison of when rule-based logic is preferable versus when a risk-aware learned policy might be worth the training overhead would be recommended.

5. You may want to refer to Figure 1 of "Distributionally robust economic dispatch using IDM for integrated electricity-heat-gas microgrid considering wind power" to refine your structure figure. Also, I strongly recommend improving the figure resolution.

6. The PF correction results clearly show the mechanism in Table 4: PF 0.84 to 0.95, apparent power 1159.18 VA to 677.20 VA, and real power 282.90 W to 278.40 W. Please emphasize earlier that PF correction mainly frees inverter capacity and reduces current, and it does not by itself produce large real energy savings.

7. The experimental load shedding case shows the main real power reduction, from 278.40 W to 213.10 W, reported as 23.46%. It would help to define what portion of the site load is “non-critical” in kW and how the critical vs non-critical classification was validated against operational constraints.

8. The scalability discussion says decision time per node stays constant and suggests edge computing due to observed latency. Since the actions are time-sensitive, please add at least one measured communication timing number, such as MQTT update rate or command round-trip delay, to support the practicality of remote control during peaks.

Reviewer #2: This manuscript presents an IoT-based demand-side energy management system for an industrial solar microgrid, integrating real-time monitoring, power factor correction, and priority-based load shedding. The system is validated through simulation and experimental results from an industrial case study. The paper addresses a practical problem; however, several aspects require further clarification and strengthening before the contribution can be fully assessed. My comments are as follows.

1.The manuscript would benefit from a clearer explanation of how IoT functionality contributes to system performance beyond monitoring and remote control. A more explicit discussion of what decisions rely on IoT data and how communication latency or data granularity affects control outcomes would strengthen the presentation.

2.The control logic is described at a high level. Providing clearer pseudo-code, flowcharts, or parameter sensitivity analysis (e.g., threshold selection) would improve reproducibility and understanding.

3.Since power factor correction plays a major role in the reported improvements, the authors are encouraged to quantify its impact separately from that of load shedding and scheduling, to better illustrate the contribution of each module.

4.Additional experiments or scenarios (e.g., different load profiles or peak periods) would help demonstrate the robustness of the proposed system. A brief comparison with baseline or commonly used DSM strategies would further strengthen the evaluation.

In summary, the paper presents a practical implementation with potential applicability. However, revisions are needed to clarify the system design choices and better support the claimed benefits. I therefore recommend major revision.

.

Reviewer #1: No

Reviewer #2: No

---

## [Author Response · Author response to Decision Letter 1]

23 Mar 2026

RESPONSES TO ACADEMIC EDITOR AND REVIEWERS’ COMMENTS

Academic editor

Thank you for your suggestion the manuscript is revised to meet PLOS ONE style requirement.

2. Please note that PLOS One has specific guidelines on code sharing for submissions in which author-generated code underpins the findings in the manuscript. In these cases, we expect all author-generated code to be made available without restrictions upon publication of the work.

Thank you for your comment, the code will be available without restrictions upon publication of the work.

3. We note that your Data Availability Statement is currently as follows:

“All relevant data are within the manuscript and its Supporting Information files.”

Thank you for a comment, all relevant data are within the manuscript and the values used to build graphs are uploaded as supporting information files.

4. PLOS requires an ORCID iD for the corresponding author in Editorial Manager on papers submitted after December 6th, 2016. Please ensure that you have an ORCID iD and that it is validated in Editorial Manager. To do this, go to ‘Update my Information’ (in the upper left-hand corner of the main menu), and click on the Fetch/Validate link next to the ORCID field. This will take you to the ORCID site and allow you to create a new iD or authenticate a pre-existing iD in Editorial Manager

We appreciate your feedback, an ORCID ID for the corresponding author is provided.

5. Please amend either the title on the online submission form (via Edit Submission) or the title in the manuscript so that they are identical.

Thank you for your comment, the title was amended on the online submission form.

6. Please include a caption for figures 2, 3, 4, 5, 6, 7, 8, 9, 10, 11, 17, and 19.

Thank you for your comment, the caption for figures 2, 3, 4, 5, 6, 7, 8, 9, 10, 11, 17, and 19. Were included throughout the revised manuscript.

7. If the reviewer comments include a recommendation to cite specific previously published works, please review and evaluate these publications to determine whether they are relevant and should be cited. There is no requirement to cite these works unless the editor has indicated otherwise

Thank you for your observation, there is no recommendation to cite specific published works from reviewers’ comments.

Reviewer: 1

1. The separation of real, reactive, and apparent power is a good framing choice, and Eq. (15) helps explain why apparent power falls more than real power. The manuscript sometimes shifts between “peak demand reduction” and “apparent power reduction” without stating which metric is being discussed in that paragraph, especially around the reported 41.58% reduction

Thank you for your observation an automatic power factor correction reduced apparent power demand by 41.58 % due to reactive power mitigation. Real power decreased by 1.6 and 2.6 % for both hardware and simulation respectively. The result shows that power factor correction reduces current and releases inverter capacity as per page 1, 30, and 31.

2. The paper benefits from having both Proteus simulation and a physical prototype, and the ±1.5% calibration statement is helpful. It would be stronger if Section 2.3.1 included one small table or example readings showing MS2203 vs PZEM-004T agreement for voltage, current, and PF, not only the final accuracy claim.

Thank you for the suggestion, the sensors were calibrated against a digital clamp meter (MS2203). Voltage, current, and power factor measurements showed a maximum deviation of ±1.5 across the tested operating range. The results confirm agreement between the PZEM-004T sensor and the reference instrument. The readings of the electrical parameters are as shown in Table 2 provided on page 11 in the manuscript

3. The paper states “peak load is 150 kWh” during 09:30–12:30, which looks like an energy unit used as power, so please clarify whether this is peak hourly energy, peak power, or another definition, and keep it consistent with Table 3.

Thank you for your concern, Figure 9 shows the highest energy consumption of 150 kWh occurs at hour 11 and the average peak energy consumption is 143.3 kWh/h occurs at the period ranges at the operating window from 09:30 to 12:30. This value represents energy consumed over a three-hour period. It does not represent instantaneous power demand. as presented on page 15, 28, and 29 in the manuscript

4. One discussion that could strengthen the paper is to comment on whether their IoT DSEM framework could incorporate an explicit “risk limit” instead of hard thresholds, similar in spirit to CVaR-based safety constraints in robust constrained MDP formulations (see a robust safe reinforcement learning-based operation method for hybrid electric-hydrogen energy system risk-based dispatch considering dynamic efficiency characteristics of electrolysers). Also, since your controller already emphasizes stability without formal proofs, a short comparison of when rule-based logic is preferable versus when a risk-aware learned policy might be worth the training overhead would be recommended.

Thank you for your suggestions The proposed control strategy follows a rule-based deterministic structure. Fixed thresholds act as operational risk limits. This approach fits industrial systems with stable load behavior and strict production constraints. Risk-aware or learning-based policies suit systems with stochastic demand and flexible operation but require higher computational effort as per page 8 and 9

5. You may want to refer to Figure 1 of "Distributionally robust economic dispatch using IDM for integrated electricity-heat-gas microgrid considering wind power" to refine your structure figure. Also, I strongly recommend improving the figure resolution.

Thank you for your suggestion since the distributionally robust economic dispatch using an integrated dispatch model for an electricity–heat–gas microgrid considers the coupling among multiple energy carriers our study concentrate only in solar microgrids. Therefore, this may be considered for future research.

6. The PF correction results clearly show the mechanism in Table 4: PF 0.84 to 0.95, apparent power 1159.18 VA to 677.20 VA, and real power 282.90 W to 278.40 W. Please emphasize earlier that PF correction mainly frees inverter capacity and reduces current, and it does not by itself produce large real energy savings.

We appreciate your comment an automatic power factor correction reduced apparent power demand by 41.58 % due to reactive power mitigation. Real power decreased by 1.6 and 2.6 % for both hardware and simulation respectively. The result shows that power factor correction reduces current and releases inverter capacity as per page 1, 30, and 31in the manuscript.

7. The experimental load shedding case shows the main real power reduction, from 278.40 W to 213.10 W, reported as 23.46%. It would help to define what portion of the site load is “non-critical” in kW and how the critical vs non-critical classification was validated against operational constraints.

Thank you for the suggestion an automated shedding of non-critical loads reduced real power demand by 23.46% during peak periods. The non-critical loads include auxiliary lighting and support equipment rated at approximately 0.063 kW. These loads represent 4 percent of the monitored site load. Classification followed production continuity, safety requirements, and operational redundancy as per page 1

8. The scalability discussion says decision time per node stays constant and suggests edge computing due to observed latency. Since the actions are time-sensitive, please add at least one measured communication timing number, such as MQTT update rate or command round-trip delay, to support the practicality of remote control during peaks

Thank you for the observation, the cloud integration improved system accessibility but introduced communication latency during remote operation. Local threshold-based control continued to execute at the controller level and remained unaffected by these delays. Communication latency influenced remote monitoring and command transmission. This observation supports future integration of edge-level processing to reduce dependence on cloud communication for time-sensitive control tasks as detailed in revised manuscript pages 9, 10 and 37.

Reviewer: 2

1. The manuscript would benefit from a clearer explanation of how IoT functionality contributes to system performance beyond monitoring and remote control. A more explicit discussion of what decisions rely on IoT data and how communication latency or data granularity affects control outcomes would strengthen the presentation

Thank you for your observation, during experimental operation, the system transmitted MQTT data updates at fixed sampling intervals, while the controller executed control actions locally upon threshold violation. The study did not target detailed network latency benchmarking as detailed in revised manuscript page 9 and 10.

2. The control logic is described at a high level. Providing clearer pseudo-code, flowcharts, or parameter sensitivity analysis (e.g., threshold selection) would improve reproducibility and understanding.

Thank you for your suggestion the control logic applies deterministic thresholds to evaluate power factor and real power in real time. The controller switches capacitor banks only after consecutive power factor readings remain below 0.95 for a defined observation window. The controller disconnects non-critical loads when real power exceeds the defined peak threshold Page 8.

3. Since power factor correction plays a major role in the reported improvements, the authors are encouraged to quantify its impact separately from that of load shedding and scheduling, to better illustrate the contribution of each module.

Thank you for your concern an automated shedding of non-critical loads reduced real power demand by 23.46% during peak periods. The non-critical loads include auxiliary lighting and support equipment rated at approximately 0.063 kW. These loads represent 4 percent of the monitored site load. Classification followed production continuity, safety requirements, and operational redundancy as per page 1

4. Additional experiments or scenarios (e.g., different load profiles or peak periods) would help demonstrate the robustness of the proposed system. A brief comparison with baseline or commonly used DSM strategies would further strengthen the evaluation.

In summary, the paper presents a practical implementation with potential applicability. However, revisions are needed to clarify the system design choices and better support the claimed benefits. I therefore recommend major revision.

Thank you for your observation additional experiments for different load profiles are recommended for future research.

---

## [Decision Letter · Decision Letter 1]

10 Apr 2026

IoT-Based Demand-Side Energy Management: Enhancing Peak Hour Efficiency Through Automated Control of Appliances

PONE-D-26-02719R1

Dear Dr. Kivevele,

We’re pleased to inform you that your manuscript has been judged scientifically suitable for publication and will be formally accepted for publication once it meets all outstanding technical requirements.

Kind regards,

Zhengmao Li

Academic Editor

PLOS One

Additional Editor Comments (optional):

Reviewers' comments:

Reviewer's Responses to Questions

**Comments to the Author**

Reviewer #1: All comments have been addressed

Reviewer #2: All comments have been addressed

2. Is the manuscript technically sound, and do the data support the conclusions?

Reviewer #1: Yes

Reviewer #2: Yes

3. Has the statistical analysis been performed appropriately and rigorously?

Reviewer #1: Yes

Reviewer #2: N/A

4. Have the authors made all data underlying the findings in their manuscript fully available?

Reviewer #1: Yes

Reviewer #2: Yes

5. Is the manuscript presented in an intelligible fashion and written in standard English?

Reviewer #1: Yes

Reviewer #2: Yes

Reviewer #1: Dear Authors, thank you for your efforts in addressing my comments. My concerns have been well handled.

Reviewer #2: (No Response)

.

Reviewer #1: No

Reviewer #2: No

---

## [Editor Report · Acceptance letter]

PONE-D-26-02719R1

PLOS One

Dear Dr. Kivevele,

I'm pleased to inform you that your manuscript has been deemed suitable for publication in PLOS One. Congratulations! Your manuscript is now being handed over to our production team.

Kind regards,

on behalf of

Dr Zhengmao Li

Academic Editor

PLOS One